# Identification of a pre-active conformation of a pentameric channel receptor

Anaïs Menny[1,2,3], Solène N Lefebvre[1,2,3], Philipp AM Schmidpeter[4], Emmanuelle Drège[5], Zaineb Fourati[6,7], Marc Delarue[6,7], Stuart J Edelstein[8], Crina M Nimigean[4], Delphine Joseph[5], Pierre-Jean Corringer[1,2]*

[1]Channel Receptors Unit, Institut Pasteur, Paris, France; [2]Unité Mixte de Recherche 3571, Centre National de la Recherche Scientifique, Paris, France; [3]Université Pierre et Marie Curie, Cellule Pasteur, Paris, France; [4]Departments of Anesthesiology, Physiology and Biophysics, Biochemistry, Weill Cornell Medicine, New York, United States; [5]BioCIS, Université Paris-Sud, CNRS, Université Paris-Saclay, Châtenay-Malabry, France; [6]Unité de Dynamique Structurale des Macromolécules, Institut Pasteur, Paris, France; [7]Unité Mixte de Recherche 3528, Centre National de la Recherche Scientifique, Paris, France; [8]Biologie Cellulaire de la Synapse, Institute of Biology, Ecole Normale Supérieure, Paris, France

**Abstract** Pentameric ligand-gated ion channels (pLGICs) mediate fast chemical signaling through global allosteric transitions. Despite the existence of several high-resolution structures of pLGICs, their dynamical properties remain elusive. Using the proton-gated channel GLIC, we engineered multiple fluorescent reporters, each incorporating a bimane and a tryptophan/tyrosine, whose close distance causes fluorescence quenching. We show that proton application causes a global compaction of the extracellular subunit interface, coupled to an outward motion of the M2-M3 loop near the channel gate. These movements are highly similar in lipid vesicles and detergent micelles. These reorganizations are essentially completed within 2 ms and occur without channel opening at low proton concentration, indicating that they report a pre-active intermediate state in the transition pathway toward activation. This provides a template to investigate the gating of eukaryotic neurotransmitter receptors, for which intermediate states also participate in activation.

*For correspondence: pjcorrin@pasteur.fr

**Competing interests:** The authors declare that no competing interests exist.

## Introduction

Pentameric ligand-gated ion channels (pLGICs) mediate fast chemical signaling between cells. Prominent members of the family include nicotinic acetylcholine (nAChRs), serotonin-type-3 (5-HT$_3$Rs), glycine (GlyRs) and γ-aminobutyric acid-A (GABA$_A$Rs) receptors that are widely expressed in the nervous system. Their mutation can cause congenital myasthenia, epilepsy, hyperekplexia and possibly autistic and schizophrenic syndromes (*Steinlein, 2012*). They are the target of major classes of therapeutic substances, including anxiolytics and sedatives, general anesthetics, smoking cessation drugs and antiemetics (*Corringer et al., 2012*). Hence, understanding the molecular mechanisms underlying their chemo-electric conversion is currently the matter of intensive work.

Following the medium resolution structure of the *Torpedo marmorata* nAChR by electron microscopy in lipids (*Miyazawa et al., 2003*), the first X-ray structures of full-length pLGICs were obtained on the bacterial homologs *Erwinia chrysanthemi* Ligand-gated Ion Channel (ELIC) (*Hilf and Dutzler, 2008*) and *Gloeobacter violaceus* Ligand-gated Ion Channel (GLIC) (*Bocquet et al., 2009*; *Hilf and Dutzler, 2009*; *Sauguet et al., 2014*), followed by structures of eukaryotic members: the GluClαR (*Hibbs and Gouaux, 2011*; *Althoff et al., 2014*), the β3GABA$_A$R (*Miller et al., 2014*), the 5-HT$_3$R (*Hassaine et al., 2014*), the α1GlyR (*Du et al., 2015*) by electron microscopy, the α$_3$GlyR

**eLife digest** In the nervous system, proteins of the pLGIC family are found in the membrane that surrounds each neuron. These proteins have channels that can allow ions to pass through the membrane and are responsible for transmitting electrical signals from one neuron to the next. Small molecules called neurotransmitters interact with the pLGICs to open or close the ion channel. If the ability of the pLGIC channels to open is altered, it can lead to behavioral changes like addiction, or diseases such as schizophrenia or epilepsy.

For a pLGIC channel to switch between the "open" and "closed" states, specific parts of the protein need to move in relation to each other. However, to study these transitions researchers have previously relied on comparing the three-dimensional structures of open and closed pLGICs extracted out of the cell membrane. Different techniques are needed to directly follow these movements within membranes.

Bacteria also have proteins belonging to the pLGIC family, and Menny et al. have now investigated one such bacterial protein to understand how pLGICs open. First, a small fluorescent molecule that glows differently if the environment around it changes was attached to various parts of the bacterial channel. These fluorescent markers revealed how several parts of the protein move and they also made it possible to measure how quickly these movements take place. Some of these movements happen before the channel opens, suggesting that the activation of this pLGIC protein happens in stages and involves the protein adopting a temporary intermediate state.

The next step will be to better understand the structure of the intermediate state, which could help us to understand how pLGICs work in the nervous systems of animals. In future this may aid the design of new drugs that can modify the activity of these channels in patients with neurological conditions or addictions.

(*Huang et al., 2015*) by crystallography and the $\alpha4\beta2$ nAChR (*Morales-Perez et al., 2016*) showing a highly conserved fold from bacteria to mammals. pLGICs are composed of five identical or homologous subunits arranged pseudosymmetrically around a central ion-conducting channel. They bind neurotransmitters within their extracellular domain (ECD), promoting remote opening of an intrinsic ion channel within their transmembrane domain (TMD). Each subunit's ECD is composed of a rigid $\beta$-sandwich. The orthosteric binding sites for neurotransmitters are located at the subunit interface, halfway between the membrane and the top of the ECD. Each subunit's TMD is composed of four membrane-spanning $\alpha$-helices named M1–4. M2 lines the ion channel, its upper part contributing to the gate that shuts the pore in the closed conformation. The well-conserved loops 2, 7 and M2–M3 are found at the ECD-TMD interface (*Figure 1A*).

pLGICs exist in multiple allosteric states. The major states can be classified as: resting (with a closed channel and a low affinity for agonists), active (with an open channel) and desensitized (with a closed channel and a high affinity for agonists) (*Changeux and Edelstein, 2005*). In addition, ensemble and single-channel kinetic analyses identified short-lived intermediate states during activation of nAChRs and GlyRs. These states, named 'flip' and 'prime', display increased agonist affinity as compared to the resting state but still carry a closed channel (*Burzomato et al., 2004*; *Lape et al., 2008*; *Mukhtasimova et al., 2009*). Finally, desensitization was early described as involving multiple conformations, including fast and slow desensitized states (*Sakmann et al., 1980*).

Among pLGICs of known structures, GLIC (*Bocquet et al., 2009*; *Prevost et al., 2012*; *Sauguet et al., 2014*), GluClα (*Hibbs and Gouaux, 2011*; *Althoff et al., 2014*) and α1GlyR (*Du et al., 2015*) were each solved in three different conformations showing multiple tertiary and quaternary conformations that are difficult to assign to a particular allosteric state. Notably, all the structures were solved on detergent-solubilized proteins, most of them constrained in a crystal lattice that appears, in several cases, to be favoring a particular conformation regardless of the nature of bound ligands (*Nury et al., 2011*; *Gonzalez-Gutierrez et al., 2012*). Complementary methods are thus required to study the conformational changes of lipid-inserted, unconstrained receptors, in a time-resolved manner.

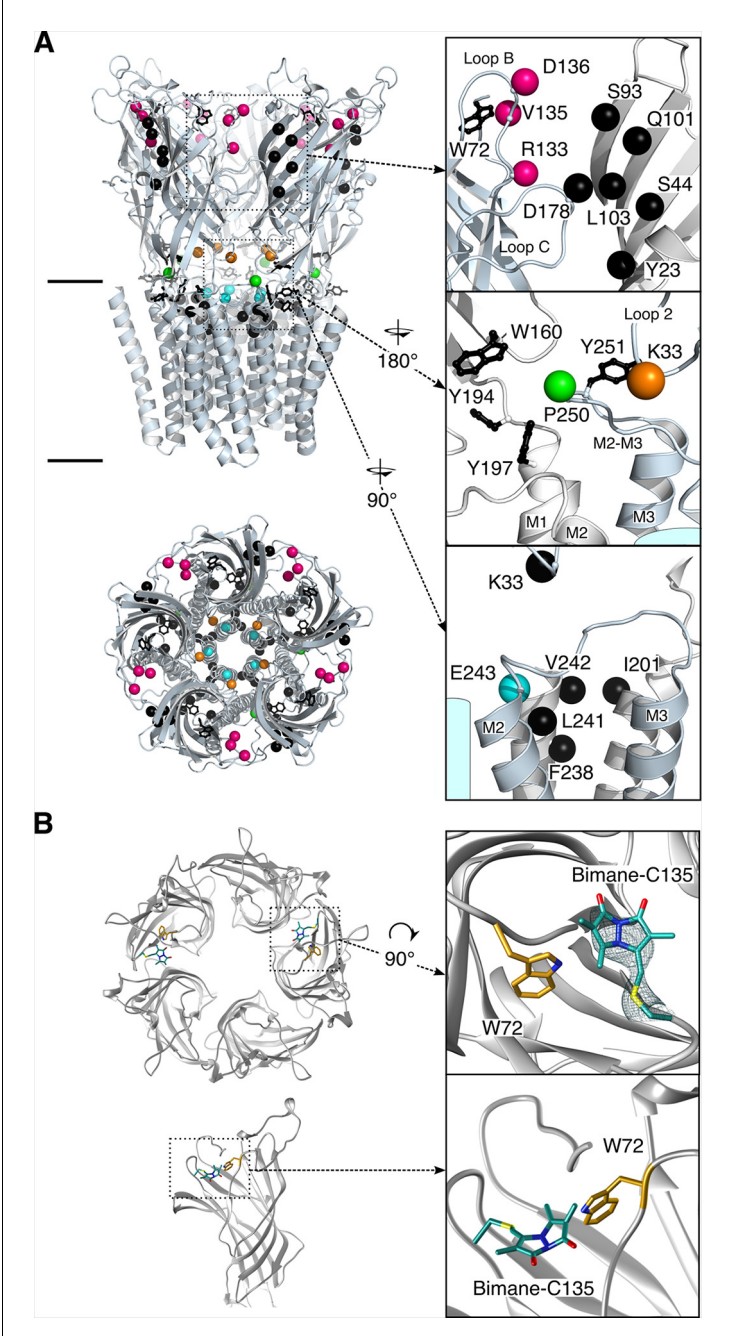

**Figure 1.** Structure of GLIC. (A) Positions for fluorophore and quencher insertion. Top panel: side view of GLIC crystalized at pH 7. Lower panel: Top-down view of GLIC. Insets show particular regions targeted for mBBr labeling. Positions where cysteines are engineered are represented in color and positions for tryptophan/tyrosine mutations are in black. All amino acids are represented by a sphere for the Cβ carbon, to show the orientation of the side chain. Endogenous tryptophans and tyrosines are in stick representation. (B) X-ray structure of the V135C-Bimane. A top and side view are presented showing the localization of the two Bimane molecules resolved in the structure. Insets show a zoom for both the top and side view of the V135C-Bimane and the putative quencher W72. In the top zoom, the experimental 2Fo-Fc density map contoured at one sigma is shown in mesh for the Bimane.

Here, we used the proton-gated ion channel GLIC (*Bocquet et al., 2007*) to identify the conformational changes that occur during gating in the family of pLGIC channels. GLIC has been crystallized in three conformations (called open, closed and locally-closed) (*Bocquet et al., 2009*; *Prevost et al., 2012*; *Sauguet et al., 2014*) and studied by molecular dynamics, normal mode analysis and EPR spectroscopy (*Sauguet et al., 2014*; *Calimet et al., 2013*; *Velisetty et al., 2014*). We used the tryptophan-induced (TrIQ) (*Mansoor et al., 2002*; *Islas and Zagotta, 2006*; *Mansoor et al., 2010*) and tyrosine-induced (TyrIQ) (*Semenova et al., 2009*; *Jones Brunette et al., 2014*) quenching methods to study short-range (5–15 Å) inter-residue, pH-elicited motions of GLIC. The methods consist in covalently linking the protein with a fluorophore, here the small and pH-insensitive bimane (*Figure 2A*), together with the insertion of a tryptophan or a tyrosine residue that quenches the fluorophore when the inter-residue Cα-Cα distances are less than approximately 15 Å and 10 Å, respectively. The Tr/TyrIQ approach was validated on the model system T4 lysosyme (*Mansoor et al., 2002*, *2010*; *Jones Brunette et al., 2014*) and was previously used to follow the allosteric transitions of proteins such as the β2-adrenoceptor (*Yao et al., 2006*) and the lactose permease (*Smirnova et al., 2014*). Previous fluorescence-based studies on pLGICs have used large fluorophores to follow conformational changes (*Talwar and Lynch, 2015*), but the TrIQ method has the unique advantage of allowing the assignment of fluorescence changes to relative motions between a fluorophore and a quencher of small sizes (*Yao et al., 2006*; *Smirnova et al., 2014*).

## Results

### Bimane labeling strategy

We focused our analysis on regions known to contribute critically to allosteric transitions of pLGICs and that are solvent-accessible, starting from the most membrane-distal region (top) of the ECD. A bimane was introduced at: (1) the subunit interface at the top of the ECD (D136 and V135 on loop B) and the middle of the ECD (R133 on loop B, at the level of the orthosteric site), to monitor the quaternary reorganizations of this domain; and (2) at both sides of the ECD-TMD interface, at the bottom of the ECD (K33 on loop 2), on the M2-M3 loop (the most N-ter proline of the loop P250), and at the top of pore-lining M2 α-helix (E243, also termed E19', which is nearby the channel gate residues I233/I9' and I240/I16') (*Figure 1A*). For each targeted position, a cysteine was engineered on the GLIC Cys-less background (C27S, which does not produce functional alterations [*Table 1*]), followed by bimane labeling through cysteine modification. Each mutant was first characterized by two-electrode voltage clamp electrophysiology in *Xenopus* oocytes before and after bimane labeling to verify the functionality of the channel. Mutants were then expressed and purified from *E. coli* membranes, labeled with bimane and studied by fluorescence in detergent (n-Dodecyl β-D-maltoside) or asolectin liposomes.

### Electrophysiological characterizations of the bimane-labeled mutants

We first developed a labeling procedure in *Xenopus* oocytes. Indeed, we found that the currently used mBBr (Monobromo bimane) does not efficiently label surface receptors, since the reporter mutant GLIC Q193C (pre-M1), which is fully inhibited by reaction with methyl-methanethiosulfate (MMTS) (*Figure 2B*), is neither affected nor protected from MMTS inhibition by mBBr treatment. Control experiments on chinese hamster ovary (CHO) cells expressing GLIC-D136C showed no surface labeling, but clear accumulation of the fluorophore in the cytoplasm (*Figure 2—figure supplement 1*). To decrease the hydrophobicity of the reagent, we synthesized the bimane-derived Bunte salt (BBs), introducing a negatively charged $SO_3^-$ leaving group (*Figure 2A* and *Figure 2—figure supplements 2* and *3*). After reaction with cysteine residues, the mBBr and BBs yield almost the same coupling product, with similar fluorescence and side-chain volume (*Mansoor and Farrens, 2004*) (*Figure 2A*). GLIC-expressing CHO cells show weak entry of the BBs in the cytoplasm and strong labeling at the membrane (*Figure 2—figure supplement 1*). Accordingly, BBs labeling of Q193C-expressing oocytes led to a severe loss of function indicating an efficient reaction with the surface receptors (*Figure 2C*).

Most mutants investigated herein were functional, generating robust pH-elicited currents, with few exceptions (*Table 1* and *Figure 2—figure supplements 4* and *5*). For all functional mutants, after BBs labeling, currents showed wild-type like biphasic GLIC activation kinetics, with time

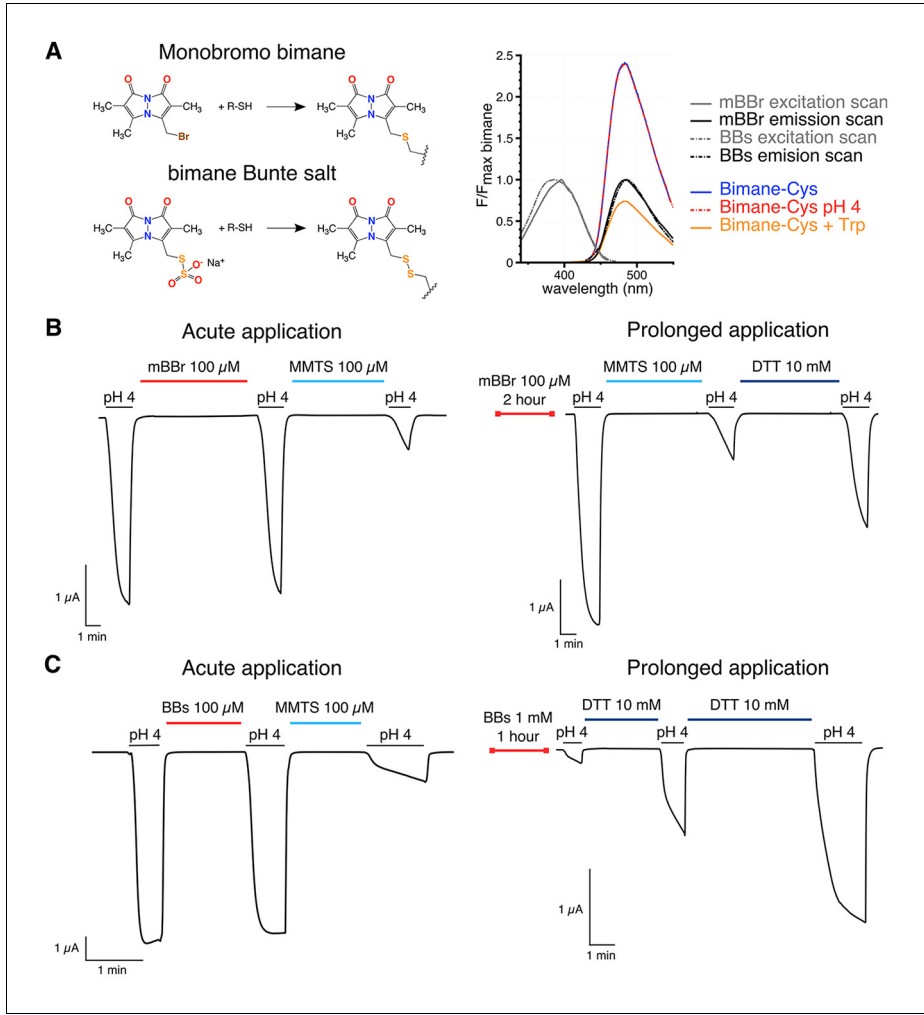

**Figure 2.** Bimanes characteristics and labeling of GLIC-expressing cells. (**A**) Left: Structures of the Monobromo bimane (mBBr) and the bimane Bunte salt (BBs) before and after reaction with cysteines. Right: Excitation and emission spectra of both fluorophores and emission spectra of the mBBr (10 μM) after reaction with cysteines (1 mM), acidification of the medium (pH 4) or addition of tryptophan (25 mM). All spectra are normalized to the peak intensity of the mBBr emission spectrum. (**B, C**) Representative electrophysiological recordings of oocytes expressing GLIC Q193C mutant. Unless otherwise indicated, oocytes were perfused with a pH 7.3 solution. (**B**) Traces showing no functional inhibition of GLIC Q193C after mBBr exposure with acute (left) or prolonged application (right), whereas the same oocytes are inhibited by reaction with MMTS, which can be reversed by application of DTT. (**C**) Left: Trace showing the inhibition of GLIC Q193C after reaction with the MMTS but not the BBs after acute application. Right: Effective inhibition of GLIC Q193C after reaction with the BBs after the oocyte was incubated for 1 hr in presence of the fluorophore, followed by reduction of the Cys-BBs bond by DTT.

The following figure supplements are available for figure 2:

**Figure supplement 1.** Bimane labeling of CHO cells.

**Figure supplement 2.** $^1$HNMR (300 MHz, $D_2O$) and $^{13}$C NMR (75 MHz, $D_2O$/MeOD) spectra of the BBs.

**Figure supplement 3.** Characterisation of the synthetized BBs.

**Figure supplement 4.** Typical electrophysiological traces for each major quenching-pairs mutants of GLIC.

**Figure supplement 5.** Immunolabeling of oocytes expressing GLIC WT and loss-of-function mutants.

**Table 1.** Dose dependence of currents and fluorescence. The table contains $pH_{50}$ and nH values obtained through Hill equation fittings of current and fluorescence dose-response curves. Imax values represent the maximal current recorded. NF stands for not functional when maximal currents are smaller than 500 nA. NA stands for non-applicable and was used for mutants that did not elicit fluorescence variations bigger than 10% across the range of pH tested. NM stands for not measured. n represents the number of experiments. For all the data, mean values are presented and error values represent the standard deviations.

| Mutant | Electrophysiology non-labeled | | | | Electrophysiology labeled | | | | Fluorescence detergent | | | Fluorescence liposomes | | |
|---|---|---|---|---|---|---|---|---|---|---|---|---|---|---|
| | pH$_{50}$ | nH | Imax | n | pH$_{50}$ | nH | Imax | n | pH$_{50}$ | nH | n | pH$_{50}$ | nH | n |
| WT | 5.3 ± 0.2 | 1.29 ± 0.06 | 6 ± 1 | 4 | NA | NA | NA | NA | NM | NM | NM | NM | NM | NM |
| C27S | 5.31 ± 0.07 | 1.6 ± 0.2 | 7.0 ± 0.4 | 5 | 5.3 ± 0.1 | 1.7 ± 0.2 | 7 ± 2 | 3 | NA | NA | 3 | NA | NA | 3 |
| R133C | 5.3 ± 0.2 | 1.8 ± 0.1 | 7 ± 3 | 4 | 5.1 ± 0.2 | 2.14 ± 0.05 | 8 ± 3 | 3 | 5.4 ± 0.3 | 1.7 ± 0.6 | 5 | 5.5 ± 0.1 | 1.3 ± 0.2 | 3 |
| R133C Y23W | 5.2 ± 0.4 | 2.0 ± 0.3 | 9.2 ± 0.8 | 3 | 5.1 ± 0.2 | 2.50 ± 0.07 | 6 ± 1 | 3 | NA | NA | 3 | NM | NM | NM |
| R133C Q101W | 4.9 ± 0.2 | 2.3 ± 0.1 | 4 ± 2 | 3 | 5.0 ± 0.1 | 2.49 ± 0.03 | 4.9 ± 0.9 | 3 | NA | NA | 3 | NM | NM | NM |
| R133C L103W | 5.1 ± 0.1 | 2.2 ± 0.4 | 6.8 ± 0.8 | 3 | 5.23 ± 0.09 | 1.7 ± 0.2 | 6 ± 2 | 3 | 5.2 ± 0.3 | 0.7 ± 0.2 | 4 | 5.50 ± 0.04 | 0.78 ± 0.07 | 3 |
| V135C W72 | 5.6 ± 0.2 | 1.7 ± 0.3 | 7.0 ± 0.9 | 4 | 5.29 ± 0.08 | 2.0 ± 0.1 | 7.2 ± 0.7 | 3 | 6.31 ± 0.09 | 1.7 ± 0.2 | 7 | 6.05 ± 0.02 | 3 ± 1 | 4 |
| V135C E67Q E75Q D91N | 5.1 | 1.8 | 7.3 | 1 | 5.1 ± 0.2 | 2.0 ± 0.4 | 6.5 ± 1.0 | 3 | 6.14 ± 0.03 | 1.8 ± 0.2 | 3 | NM | NM | NM |
| D136C | 5.3 ± 0.1 | 1.9 ± 0.2 | 7.0 ± 0.9 | 4 | 5.29 ± 0.04 | 2.0 ± 0.3 | 8.3 ± 0.9 | 3 | NA | NA | 3 | NA | NA | 3 |
| D136C S93W | 4.50 ± 0.05 | 2.1 ± 0.4 | 3 ± 1 | 3 | 4.60 ± 0.06 | 1.5 ± 0.1 | 2.1 ± 0.9 | 3 | NA | NA | 4 | NM | NM | NM |
| D136C Q101W | 5.17 ± 0.08 | 2.3 ± 0.1 | 8 ± 2 | 3 | 5.4 ± 0.2 | 2.2 ± 0.3 | 9 ± 3 | 3 | 5.85 ± 0.08 | 0.96 ± 0.08 | 4 | 5.8 ± 0.1 | 1.3 ± 0.6 | 3 |
| D136C D178W | 5 | 1.8 | 6.2 | 1 | 5.3 | 1.8 | 6.6 | 1 | NA | NA | 3 | NM | NM | NM |
| K33C W160 | 5.88 ± 0.03 | 1.8 ± 0.1 | 9 ± 2 | 3 | 5.6 ± 0.2 | 1.8 ± 0.4 | 8 ± 2 | 3 | 6.22 ± 0.03 | 2.0 ± 0.4 | 4 | 5.95 ± 0.07 | 1.25 ± 0.09 | 6 |
| K33C W160F | 4.55 ± 0.03 | 2.6 ± 0.2 | 3.0 ± 0.1 | 3 | 4.4 ± 0.1 | 2.2 ± 0.6 | 1 ± 1 | 4 | NA | NA | 3 | NA | NA | 3 |
| P250C Y197 | 5.1 ± 0.1 | 1.54 ± 0.09 | 7.0 ± 1.2 | 3 | 4.8 ± 0.2 | 2.3 ± 0.4 | 4.3 ± 0.8 | 3 | 5.88 ± 0.09 | 2.1 ± 0.8 | 5 | 5.5 ± 0.3 | 2.2 ± 0.6 | 4 |
| P250C W160F | NF | NF | NF | 3 | NF | NF | NF | 4 | 6.06 ± 0.06 | 1.9 ± 0.1 | 3 | NM | NM | NM |
| P250C Y194F | 5.1 ± 0.2 | 1.7 ± 0.4 | 6.4 ± 0.8 | 3 | 4.5 ± 0.2 | 1.8 ± 0.2 | 2 ± 1 | 3 | 5.97 ± 0.02 | 1.4 ± 0.4 | 4 | NM | NM | NM |
| P250C Y197F | 5.2 ± 0.1 | 1.3 ± 0.3 | 4.1 ± 0.1 | 5 | 4.73 ± 0.09 | 1.2 ± 0.2 | 1.5 ± 0.9 | 3 | NA | NA | 3 | NA | NA | 3 |
| P250C Y251F | NF | NF | NF | 7 | NF | NF | NF | 3 | NA | NA | 4 | NM | NM | NM |
| E243C | 5.1 ± 0.1 | 1.7 ± 0.3 | 8 ± 1 | 4 | 4.7 ± 0.2 | 1.7 ± 0.3 | 5 ± 4 | 3 | 4.5 ± 0.2 | 2.0 ± 0.2 | 4 | 4.8 ± 0.1 | 0.9 ± 0.3 | 6 |
| E243C K33W | NF | NF | NF | 4 | NF | NF | NF | 3 | NA | NA | 3 | NM | NM | NM |
| E243C I201W | NF | NF | NF | 10 | NF | NF | NF | 3 | 5.67 ± 0.04 | 1.549 ± 0.004 | 3 | 5.8 | 1.4 | 2 |
| E243C F238W | 5.2 ± 0.2 | 1.4 ± 0.2 | 7 ± 1 | 4 | 4.7 ± 0.3 | 1.5 ± 0.5 | 5 ± 3 | 3 | NA | NA | 3 | NM | NM | NM |
| E243C L241W | 5.7 ± 0.2 | 0.69 ± 0.06 | 5.09 ± 0.08 | 3 | 4.8 ± 0.1 | 0.61 ± 0.03 | 2 ± 1 | 3 | NA | NA | 3 | NM | NM | NM |
| E243C V242W | NF | NF | NF | 4 | NF | NF | NF | 5 | NA | NA | 3 | NM | NM | NM |

constants ranging from 1 to 2.8 s (τ1) and 5 to 8.9 s (τ2) (*Table 2* and *Figure 2—figure supplement 4*), except for P250C baring its endogenous quencher Y197 (termed P250C-Y197) which displays slower kinetics of activation (5.4 s and 50 s). pH-current relationship measurements show ΔpH$_{50}$ (pH value for which half of the maximal electrophysiological response is recorded) of less than one between wild-type and BBs-labeled mutants (*Table 1*). Thus, the labeling with the small BBs probe weakly affects the gating in most cases.

## Analysis of bimane-quencher pairs in the ECD in DDM micelles

Emission spectra of DDM-solubilized mutants labeled with bimane were recorded under steady state conditions (30 s post proton-application, excitation at 385 nm [see Materials and methods]). Fluorescence intensities were measured at the emission peak at various pH values (from pH 7.3 to pH 3) (*Figure 3—figure supplement 2*) and normalized to the intensity of the respective bimane-mutant under denaturing conditions (1% SDS). GLIC being activated by protons, we additionally performed control experiments to confirm that, as previous studies have shown (*Jones Brunette et al., 2014*), both the bimane fluorescence and its quenching by tryptophans are unaffected by proton concentrations ranging from pH 9 to pH 2 (*Figure 3—figure supplement 1A,B*) (see Materials and methods). Hence, the fluorescence variations of bimane-labeled mutants can be interpreted as reporting local structural reorganizations.

When the bimane is introduced on loop B (top of the ECD) at positions 133 or 136 (termed Bimane-133 and Bimane-136 [*Figure 1A*]), the fluorescence shows little variation in the pH 7.3–3 range, indicating that nearby putative quenching residues have a weak impact on the pH-dependent bimane fluorescence (*Figure 3A,B*). Bimane-133 shows a marked blue shift (15 nm) and a fluorescence two times higher than the denatured protein at pH 7.3 possibly indicating the probe when reacted to position 133 is located in a confined/hydrophobic environment (*Kosower et al., 1982*; *Skjold-Jørgensen et al., 2015*) (*Figure 3A* and *Figure 3—figure supplement 2*). To generate quenching pairs, we introduced tryptophan residues on the adjacent subunit β-sandwich (*Figure 1A*). For mutant Bimane-133, the introduction of W23 (β1 strand), W44 (β2 strand) and W101 (β6 strand) has little impact on the fluorescence at all pHs suggesting that these positions are never within quenching distances of the bimane (*Figure 3A*). For mutant Bimane-136, the introduction of W93 (β5 strand) produces a strong fluorescence decrease at all pHs indicating in this case that this residue is always within quenching distance of the bimane (*Figure 3B*). In contrast, both the Bimane-133-W103 and Bimane-136-W101 show a marked pH-dependent fluorescence decrease (*Figure 3A,B*). These data show that the residues at positions 133 and 103, and those at positions 136 and 101 come closer upon acidification of the receptors.

When the bimane is introduced at position 135 (loop B), the pH-fluorescence relationship curve shows an inverted bell-shape, starting after normalization at 0.7, decreasing down to 0.3 at pH 5–4, then increasing up to 0.6 at pH 3 (*Figure 3C*). The GLIC structures show a tryptophan (W72 on loop 4) on the same subunit that could play the role of an endogenous quencher (*Figure 1A*). W72 is a residue strictly conserved in all pLGICs (*Corringer et al., 2012*). To determine whether W72 has a role in the fluorescence phenotype while avoiding severe mutation-induced structural alteration that

**Table 2.** Activation kinetics of GLIC mutants labeled with bimane. The τ values were obtained through a double exponential fit to the electrophysiology traces (see Materials and methods). n represents the number of recordings used for the analysis. For all the data, mean values are presented and error values represent the standard deviations.

| Mutant | τ1 (s) | τ2 (s) | τw (s) | n |
|---|---|---|---|---|
| R133C L103W | 1.0 ± 0.5 | 6 ± 5 | 3 ± 2 | 3 |
| V135C W72 | 1.9 ± 0.4 | 9 ± 5 | 3 ± 1 | 3 |
| D136C Q101W | 0.9 ± 0.2 | 5 ± 4 | 2 ± 2 | 3 |
| K33C W160 | 1.7 ± 0.6 | 7 ± 5 | 5 ± 3 | 3 |
| P250C Y197 | 5.4 ± 0.7 | 50 ± 8 | 30 ± 10 | 3 |
| E243C | 2.80 ± 0.09 | 8.9 ± 0.6 | 3.57 ± 0.06 | 3 |

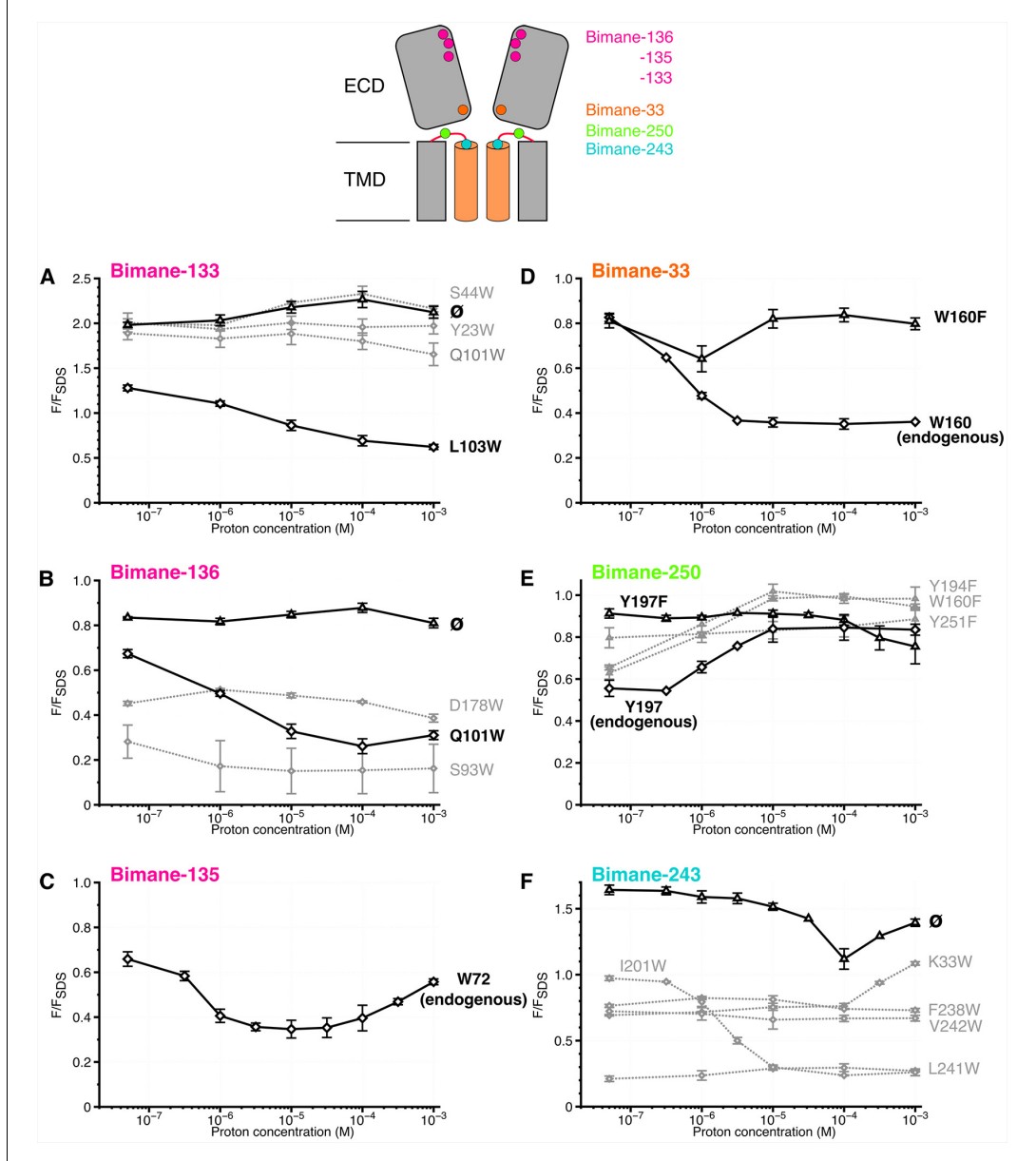

**Figure 3.** Steady-state variations of fluorescence of Bimane-GLIC mutants in detergent. Top: Cartoon view of two subunits of GLIC, all the sites of bimane labeling are represented by a colored sphere. (A, B, C, D, E, F): The data show the fluorescence peak values as a function of the proton concentration, normalized to the peak intensity after SDS treatment. The same color code as the top cartoon was used to indicate the site of bimane labeling for each graph. Within one graph, the major quenching pairs with or without quencher are shown in bold black caption and secondary tryptophan quenchers are captioned in grey. The symbol ø was used for recordings made on receptors baring no other mutation than the one for bimane labeling and for which no endogenous quenchers were identified; otherwise the later are indicated in bold black caption. For all the points, mean values are presented and error bars represent the standard deviations.

The following figure supplements are available for figure 3:

**Figure supplement 1.** Spectral characteristics of mBBr and Bimane-GLIC mutants.

**Figure supplement 2.** Emission spectra of Bimane-GLIC mutants in detergent.

would result from the loss of W72, we solved the X-ray structure of Bimane-135 GLIC at pH 4 at 2.6 Å resolution (*Figure 1B* and *Supplementary file 1A*). The structure shows a protein conformation quasi-identical to the wild-type GLIC structure (*Bocquet et al., 2009*) at pH 4 (RMSD = 0.26 Å), with an additional electron density around C135 allowing unambiguous construction of a bimane moiety in two out of the five subunits. The data indeed show that the bimane is at a minimal 3.2 Å distance from the indole ring of W72, which is therefore most likely the major partner in the pH-elicited fluorescence changes of Bimane-135 (*Figure 1B*). As the 135 side chain projects at the subunit interface, the quenching of Bimane-135 by W72 seen up to pH 4 suggests a contraction of the domains interfaces, driving the bimane moiety toward W72, although other mechanisms are possible. At very low pH, the bimane appears to move away from the W72.

Finally, we investigated a local tertiary motion of the orthosteric site, by combining the Bimane-136 (loop B) mutation with the introduction of a tryptophan at position 178 at the tip of the loop C from the same subunit (*Figure 1A*). W178 leads to a 50% decrease in fluorescence at all pHs tested, indicating that both positions (136 and 178) are within quenching distance, and suggesting that their relative distance is unchanged upon increase of proton concentration (*Figure 3B*), in agreement with EPR measurements that suggest an immobility of this position upon pH drop (*Velisetty and Chakrapani, 2012*). It is noteworthy that loop C has been proposed to undergo a closing motion during activation of pLGICs, a feature not seen here with fluorescence.

## Analysis of bimane-quencher pairs at the ECD/TMD interface in DDM micelles

First, we introduced a bimane at the tip of loop 2 in the ECD (Bimane-33), which lies on top of the M2 helix (*Figure 1A*). Bimane-33 shows a twofold pH-dependent decrease in fluorescence from pH 7.3 to pH 5 (*Figure 3D*). A putative quencher near Bimane-33, W160, is located on the adjacent subunit on the β9 strand at the bottom of the outer beta sandwich (*Figure 1A*). Its mutation (W160F) essentially abolishes the proton-elicited fluorescent decrease, with only a 20% decrease in intensity at pH 6, followed by a return to pH 7-like values at lower pHs, showing that Bimane-33 and W160 come closer together in the presence of protons.

Second, we introduced a bimane at position 250 within the M2-M3 loop (*Figure 1A*). Bimane-250 shows a 30% increase in intensity as the pH is decreased from 7.3 to 3 (*Figure 3E*). The effect of three surrounding tyrosines and a single tryptophan were investigated through mutation to phenylalanine (*Figure 1A*). Mutants Bimane-250-W160F (β9 strand) and -Y194F (pre-M1) both lead to a modest increase in fluorescence at all pHs suggesting that W160 and Y194 remain at similar quenching distances of the bimane, independently of the protein conformation (*Figure 3E*). In contrast, Y197F (M1 helix) and Y251F (M2-M3 loop) reduce the fluorescence quenching at high pHs, yielding a flat pH-dependent curve. The mutant Bimane-250-Y251F is non-functional (*Table 1*), which could result in its incapacity to visit different conformations and thus account for this phenotype. In contrast, the mutant Bimane-250-Y197F is functional, the unquenching phenotype showing that positions 250 and 197 move away from each other in the presence of protons (*Figure 3E*).

Third, we introduced a bimane on the top of the M2 α-helices (Bimane-243) (*Figure 1A*). Upon acidification, the fluorescence decreases slightly at pH 4 and then increases at pH 3, pointing to complex reorganizations. We introduced tryptophan residues around the probe but failed to engineer new pH-dependent quenching pairs (*Figure 1A*). Indeed, the mutants were either non-functional (Bimane-243-W33, -W201 and -W242) or functional but not eliciting pH-dependent fluorescence variations (Bimane-243-W238 and -W241) (*Figure 3F* and *Table 1*). Nonetheless, regardless of the functionality of the mutants, Bimane-243 is robustly quenched by all the introduced tryptophans at pH 7.3. As most introduced tryptophans are located in the TMD, the fluorescence data, in combination with GLIC structure inspection, suggests that the bimane fused bis-heterocycle at position 243 points toward the helix bundle of the adjacent subunit. In addition, the mutant Bimane-243 shows an emission blue shift (10 nm) and a fluorescence 1.6 times higher than the denatured protein at pH 7.3 possibly reporting a confined/hydrophobic environment of the probe (*Figure 3F* and *Figure 3—figure supplement 2*) adding more evidence to its presence inside the bundle of helix. Hence, the unassigned fluorescence variations at position 243 could account for motions of the top of the M2-helix relative to the rest of the subunit's TMD.

## The six fluorescent sensors report allosteric reorganizations of GLIC

Altogether, the TrIQ/TyrIQ analysis generates a series of fluorescent reporters spanning from the apex of the ECD to the top of the pore lining M2 helices. Sensors at positions 136–101, 135–72, 133–103, 33–160, 250–197 and 243 follow the reorganizations of fully functional channels and have been selected for further analysis. In contrast, some sensors, such as the 250–160 and 243–101, are of non-functional channels but still show specific quenching signals. This illustrates that particular conformational motions are not necessarily linked to pore opening. Since the molecular mechanisms impairing the function of these mutants are not known, they were not used for further analysis.

At each selected position, changes in bimane/quencher distances could be the result of: (1) local side chains reorganizations, for example due to the protonation of residues surrounding the fluorophore and/or quencher, possibly affecting their orientation, regardless of the allosteric state of the receptor, or (2) global allosteric protein motions, mainly comprising backbone reorganizations. Two sets of experiments strongly support the latter hypothesis. First, we generated the mutant Bimane-135-W72-E67Q-E75Q-D91N, for which all titratable residues surrounding the Bimane-135-W72 pair were removed (*Figure 4A*). This mutant shows the same pH-dependence of fluorescence as the simple mutant Bimane-135-W72 (*Figure 4A*), supporting the hypothesis that the fluorescence variations are independent of the protonation of surrounding titratable residues. Second, we measured fluorescence variations in the presence of propofol, a negative allosteric modulator of GLIC. Propofol binds to the transmembrane domain of GLIC with an $IC_{50} \approx 25$ µM and stabilizes closed channel conformations (*Nury et al., 2011*). In presence of saturating concentrations of propofol, mutants Bimane-136-W101, Bimane-135-W72, Bimane-250-Y197 and Bimane-243 show a shift of the pH-fluorescence relationship curve toward higher proton concentrations, in agreement with the effect of an allosteric inhibitor (*Figure 4B*). The observation that the binding of an effector to the TMD influences the fluorescence variations at the top of the ECD, more than 50 Å away, establishes that the fluorescence sensors indeed report global allosteric motions.

Interestingly, the quenching pairs in the ECD, each located across the subunits' interface, were mostly introduced on rigid loops and $\beta$ strands (e.g. loop B and $\beta$6 strand, *Figure 1A*). As they all show a decrease in fluorescence at low pH, the fluorescence variations likely reflect rigid body motions of the ECD's $\beta$-sandwiches, thus coming closer to one another during allosteric transitions. In addition, the increase in fluorescence at low pH for the pair 250–197 reveals that another major allosteric reorganization of GLIC is the separation of loop M2-M3 (P250) from the top of M1 (Y197).

To link these motions to the gating transition of the receptor, we further studied GLIC in lipid bilayers, performing both steady state and real-time fluorescence measurements.

## Detergent-solubilized and lipid-reconstituted GLIC show similar pH-elicited reorganizations

To examine the impact of the membrane environment on the receptor reorganizations, we reconstituted selected bimane-GLIC mutants in asolectin liposomes, which are a mixture of lipids that were successfully used to reconstitute GLIC in a functional state (*Labriola et al., 2013*; *Velisetty and Chakrapani, 2012*).

Strikingly, at the top and middle of the ECD (Bimane-133 ± W103, Bimane-135-W72 and Bimane-136 ± W101), steady-state fluorescence variations are similar, if not identical, in detergent micelles and liposomes (*Figure 5A,B,C*, *Table 1* and *Table 3*), indicating that the corresponding movements are independent of the membrane environment.

At the ECD-TMD interface, liposome reconstitution causes a significant increase in fluorescence of Bimane-33 and Bimane-250-Y197F mutants at pH 8/7, as compared to detergent conditions (30% and 50%, respectively) (*Table 3*). To better visualize the pH-dependent changes, the data were normalized to the pH 8/7 values, showing that the marked pH-dependent changes of Bimane-250 and Bimane-33 are conserved in lipids, although Bimane-33-W160F is significantly brighter in detergent (*Figure 5D,E*). Additionally, Bimane-243 shows similar pH-dependent variations in fluorescence in detergent and lipids (*Figure 5F*). The data thus suggest that the relative movements at positions 33–160 and 250–197 are also essentially conserved in lipids.

We propose that the differences in fluorescence intensity at certain positions between detergent and lipids are in line with their proximity to the membrane. Indeed, the GLIC structure at pH 4 shows the presence of a bundle of detergent within the pore and of a bound lipid in the upper part of the

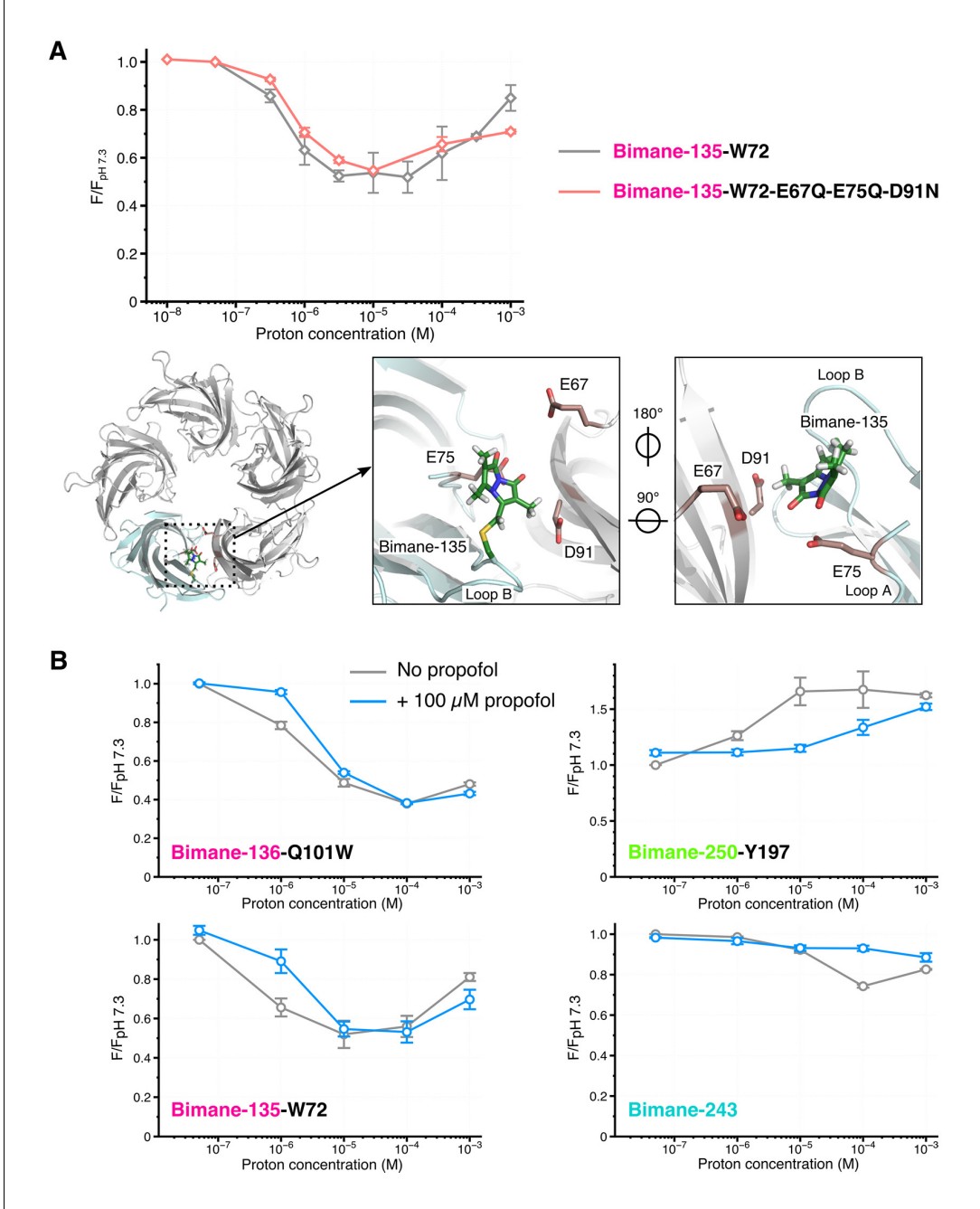

**Figure 4.** Effects of amino acid protonation and propofol on the fluorescence variations. (A) Top: Fluorescence intensities of the mutants Bimane-135-W72 and Bimane-135-W72-E67Q-E75Q-D91N normalized on each mutant's intensity at pH 7.3. Bottom, from left to right: Top view of the extracellular domain of GLIC Bimane-C135 showing one subunit in light blue, and the Bimane-C135 (green), E67, E75 and D91 (red) in stick representation; zoom on the subunits interface of the ECD, top view; zoom on the subunits interface viewed from the inside of the ECD. (B) For each pH, fluorescence recordings of Bimane-labeled mutants were first made without propofol, followed by addition of propofol at 100 μM final concentration and re-recording of the same samples. All the data were normalized on the value of fluorescence at pH 7.3 without propofol. For all the data, mean values are presented and error bars are calculated as standard deviations (n = 3 to 6).

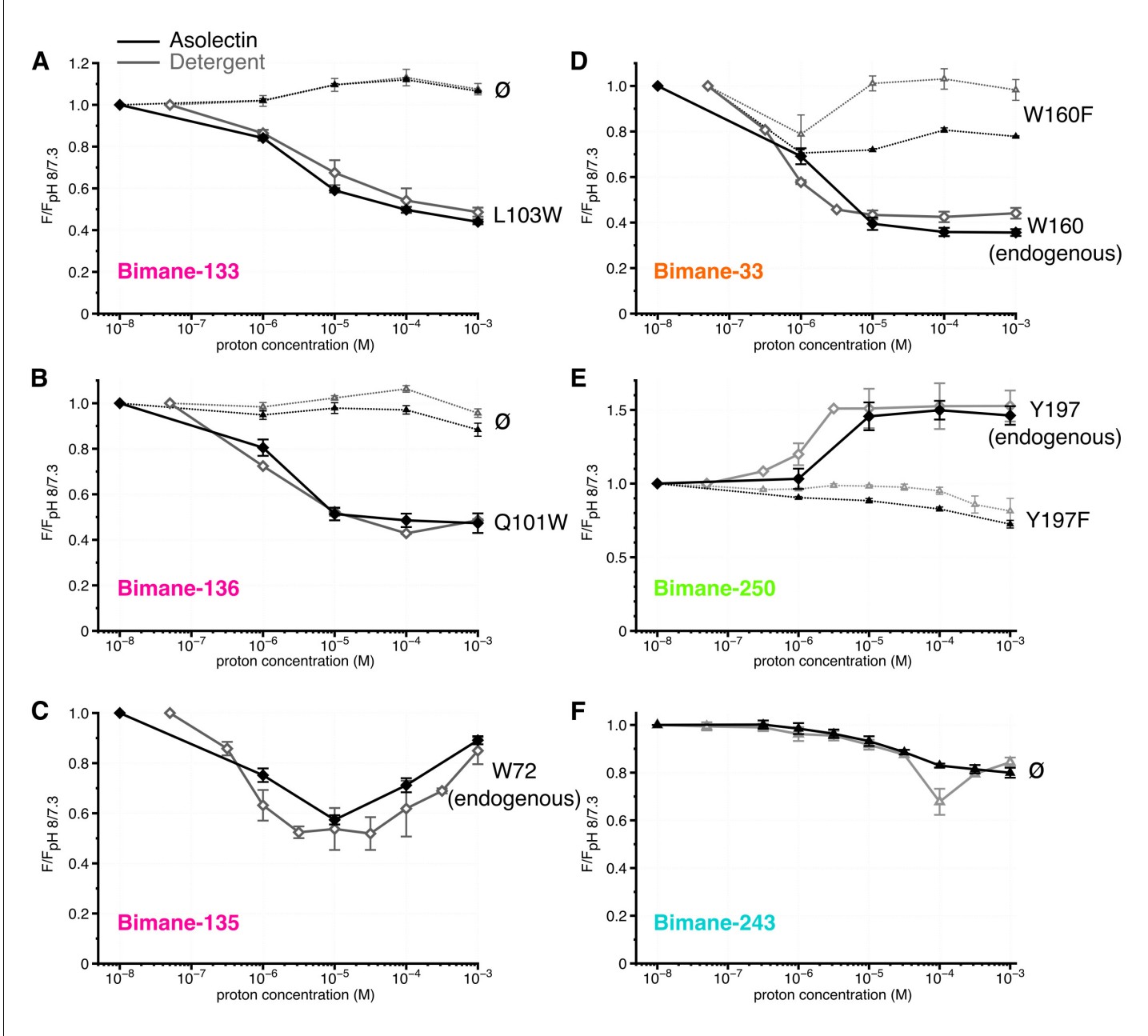

**Figure 5.** Steady-state fluorescence of Bimane-GLIC in liposomes. (A,B,C,D,E,F) The data show the fluorescence peak value for a given pH, normalized to the peak intensity of each mutant at the highest pH recorded (pH 8 or 7.3). Black lines represent the fluorescence values in liposomes and grey lines in detergent, for comparison. Plain lines represent the major fluorescent sensor and dashed lines the corresponding mutant in absence of quencher. The symbol ø was used for recordings made on receptors baring no other mutation than the one for bimane labeling and for which no endogenous quenchers were identified; otherwise the later are indicated in black caption. For all the data, mean values are presented and error bars are calculated as standard deviations.

TMD (*Bocquet et al., 2009*). These bound molecules could contribute to the differences in microenvironment observed here.

It is noteworthy that, contrary to GLIC, most pLGICs are highly sensitive to the membrane environment. For instance, CHAPS-solubilized muscle nAChRs are strongly stabilized in a desensitized conformation (*Martinez et al., 2002*) and their reconstitution in the absence of anionic lipids or cholesterol (PC liposomes) yields an 'uncoupled' conformation that binds agonist with resting state–like

**Table 3.** Fluorescence values at pH 7/8 in detergent or asolectin liposomes. The data presented are the values of fluorescence at the highest tested pH, normalized to the fluorescence after denaturation (1% SDS). n represents the number of experiments. For all the data, mean values are presented and error values represent the standard deviations.

| Mutant | $F/F_{SDS}$ in detergent | n | $F/F_{SDS}$ in asolectin | n |
|---|---|---|---|---|
| Bimane-133 | 1.98 ± 0.03 | 4 | 1.91 ± 0.02 | 3 |
| Bimane-133-W103 | 1.28 ± 0.03 | 4 | 1.182 ± 0.003 | 3 |
| Bimane-135-W72 | 0.66 ± 0.03 | 7 | 0.76 ± 0.03 | 4 |
| Bimane-136 | 0.835 ± 0.003 | 3 | 0.87 ± 0.03 | 3 |
| Bimane-136-W101 | 0.67 ± 0.02 | 4 | 0.69 ± 0.02 | 4 |
| Bimane-33-W160 | 0.83 ± 0.02 | 4 | 1.09 ± 0.03 | 6 |
| Bimane-33-W160F | 0.81 ± 0.08 | 3 | 0.87 ± 0.02 | 3 |
| Bimane-243 | 1.68 ± 0.02 | 4 | 1.60 ± 0.01 | 6 |
| Bimane-250-Y197 | 0.55 ± 0.05 | 4 | 0.60 ± 0.01 | 4 |
| Bimane-250-Y197F | 0.93 ± 0.01 | 3 | 1.42 ± 0.08 | 3 |

low affinity, but does not undergo agonist-evoked conformational transitions (*daCosta et al., 2013*). GLIC does not exhibit the same propensity to adopt an uncoupled conformation and, contrary to ELIC, retains the ability to undergo proton-elicited conformational change and to activate in PC liposomes and other lipid mixtures (*Labriola et al., 2013*; *Velisetty and Chakrapani, 2012*; *Dellisanti et al., 2013*). A cluster of Trp residues, strengthening M4 interactions with M1/M3, was found to critically contribute to this robustness of GLIC function in various lipid environments (*Hénault et al., 2015*; *Carswell et al., 2015*). Our data show similar pH-elicited movements in detergent and liposomes, suggesting that this robustness also applies to detergent-solubilized GLIC, at least at the level of the positions herein investigated.

## Rapid kinetic analyses of selected lipid-reconstituted pairs by stopped-flow

We analyzed the time-course of the conformational motions using a stopped-flow apparatus. Bimane-labeled GLIC mutants reconstituted in liposomes at pH 8 were either kept at pH 8 or mixed with acidic solutions buffering the proton concentration to final pH values of 6, 5 and 4, and followed by fluorescence. In all cases, the signal recorded 30 s after mixing was in the same range as in steady-state conditions (*Table 4*).

For most positions, the changes in fluorescence (as compared to the fluorescence at pH 8) are found to mainly occur in the dead-time of the instrument, during the first 2 ms of solution mixing (*Figure 6A*). While this very fast component is not resolved here, completion of the fluorescence variations in 2 ms indicates an upper value of the time constant below 1 ms. The subsequent variations of fluorescence were subjected to multi-exponential fits, excluding the first 3 ms of recording that show high variability (see Materials and methods and *Figure 6—figure supplement 1*). The binning analysis of all multi-exponentials allows the separation of kinetic values in three phases with time constants ranging from 5–24 ms (fast), to 24–966 ms (intermediate) and 966 ms–30 s (slow) (*Figure 6B* and *Table 5*). At positions 133, 136, 33 and 250, the dead-time 2 ms ('very fast') component, evaluated at the start of the multi-exponential fit at 5 ms, accounts for the majority of the fluorescence variation, especially at pH 4 where 72% to 90% of the fluorescence changes are completed (*Figure 6C* and *Table 5*). Therefore, the motion of positions 136–101, 133–103 and 33–160 moving closer together, and the separation of 250–197 occur with very fast kinetics.

Interestingly, Bimane-243 shows a distinctive kinetic pattern, as its very fast component accounts for only 40% of the fluorescence variation. This is followed by fast and intermediate components accounting for 30% and 20% of the fluorescence variation, respectively (*Figure 6C* and *Table 5*). Although the fluorescence changes at position 243 could not be assigned to particular

**Table 4.** Comparison of stopped-flow and steady-state fluorescence variations. The two columns show the maximal fluorescence variation recorded in the steady-state conditions or after 30 s of stopped-flow recordings. (+) indicates an increase of fluorescence and (−) indicates a decrease. n represents the number of experiments. For all the data, mean values are presented and error values represent the standard deviations.

| Mutant | pH | Maximal ΔF steady-state (%) | n | Maximal ΔF stopped-flow (%) | n |
|---|---|---|---|---|---|
| Bimane-133-W103 | pH 6 | (−) 16 ± 1 | 3 | (−) 11 ± 3 | 4 |
|  | pH 5 | (−) 41 ± 1 | 3 | (−) 28 ± 9 | 3 |
|  | pH 4 | (−) 50 ± 1 | 3 | (−) 38 ± 5 | 4 |
| Bimane-135-W72 | pH 6 | (−) 25 ± 3 | 4 | (−) 20 ± 7 | 6 |
|  | pH 5 | (−) 43 ± 2 | 4 | (−) 29.9 ± 0.9 | 6 |
|  | pH 4 | (−) 29 ± 3 | 4 | (−) 20 ± 3 | 4 |
| Bimane-136-W101 | pH 6 | (−) 20 ± 4 | 3 | (−) 12 ± 3 | 4 |
|  | pH 5 | (−) 49 ± 3 | 3 | (−) 43 ± 4 | 4 |
|  | pH 4 | (−) 53 ± 2 | 5 | (−) 45 ± 4 | 4 |
| Bimane-33-W160 | pH 6 | (−) 31 ± 4 | 6 | (−) 28 ± 5 | 3 |
|  | pH 5 | (−) 61 ± 3 | 6 | (−) 55 ± 4 | 4 |
|  | pH 4 | (−) 64 ± 2 | 6 | (−) 56 ± 3 | 3 |
| Bimane-250-Y197 | pH 6 | (+) 3 ± 7 | 4 | (+) 15 ± 8 | 4 |
|  | pH 5 | (+) 46 ± 9 | 4 | (+) 60 ± 20 | 4 |
|  | pH 4 | (+) 50 ± 6 | 4 | (+) 70 ± 40 | 4 |
| Bimane-243 | pH 6 | (−) 2 ± 2 | 6 | (−) 13 ± 10 | 4 |
|  | pH 5 | (−) 7 ± 2 | 6 | (−) 19 ± 9 | 4 |
|  | pH 4 | (−) 17.0 ± 0.8 | 6 | (−) 18 ± 9 | 4 |

conformational motions, these data suggest that the upper part of the channel moves over a broad time scale.

Finally, at all positions, slow components of fluorescence variations were recorded with time constants ranging from 1 to 30 s, particularly at positions V135 at the top of the ECD, P250 on the M2-M3 loop, and E243 on the top of the M2 α-helix (*Figure 6C*). As the slow components at all positions (except E243) show fluorescence variations in the same direction as the very fast component, it may suggest that the whole protein follows the previously described motions under prolonged applications of protons.

## Ion flux assays in liposomes and comparison with fluorescence data

We next investigated how the kinetics of the conformational changes measured above compare with kinetics of activation/desensitization of GLIC. For this, we recorded channel activity from purified GLIC reconstituted in asolectin liposomes, using a fluorescence-based sequential-mixing stopped-flow assay (*Rusinova et al., 2014*; *McCoy et al., 2014*; *Posson et al., 2015*). GLIC Cys-less and mutant Bimane-136-W101 were reconstituted in liposomes containing the water-soluble fluorophore 8-aminonaphthalene-1,3,6-trisulfonic acid (ANTS) and mixed sequentially, first with protons to activate the channels (step 1), and then with the channel-permeable ANTS-fluorescence-quencher thallium to assess the flux through open channels (step 2). By including a variable delay time (10–200 ms) between the two steps, we are able to capture the channels in various levels of activation and/or desensitization and thus populate different functional states. Upon thallium influx into the liposomes via active GLIC the ANTS fluorescence gets quenched, and the quenching rate is proportional to channel activity. Due to intrinsic variations in liposome sizes (the mean diameter of vesicles is ~ 150 nm [*Ingólfsson and Andersen, 2010*]) and different numbers of channels per liposome, the data were fit with a stretched exponential in order to obtain the average quenching rate at 2 ms, a

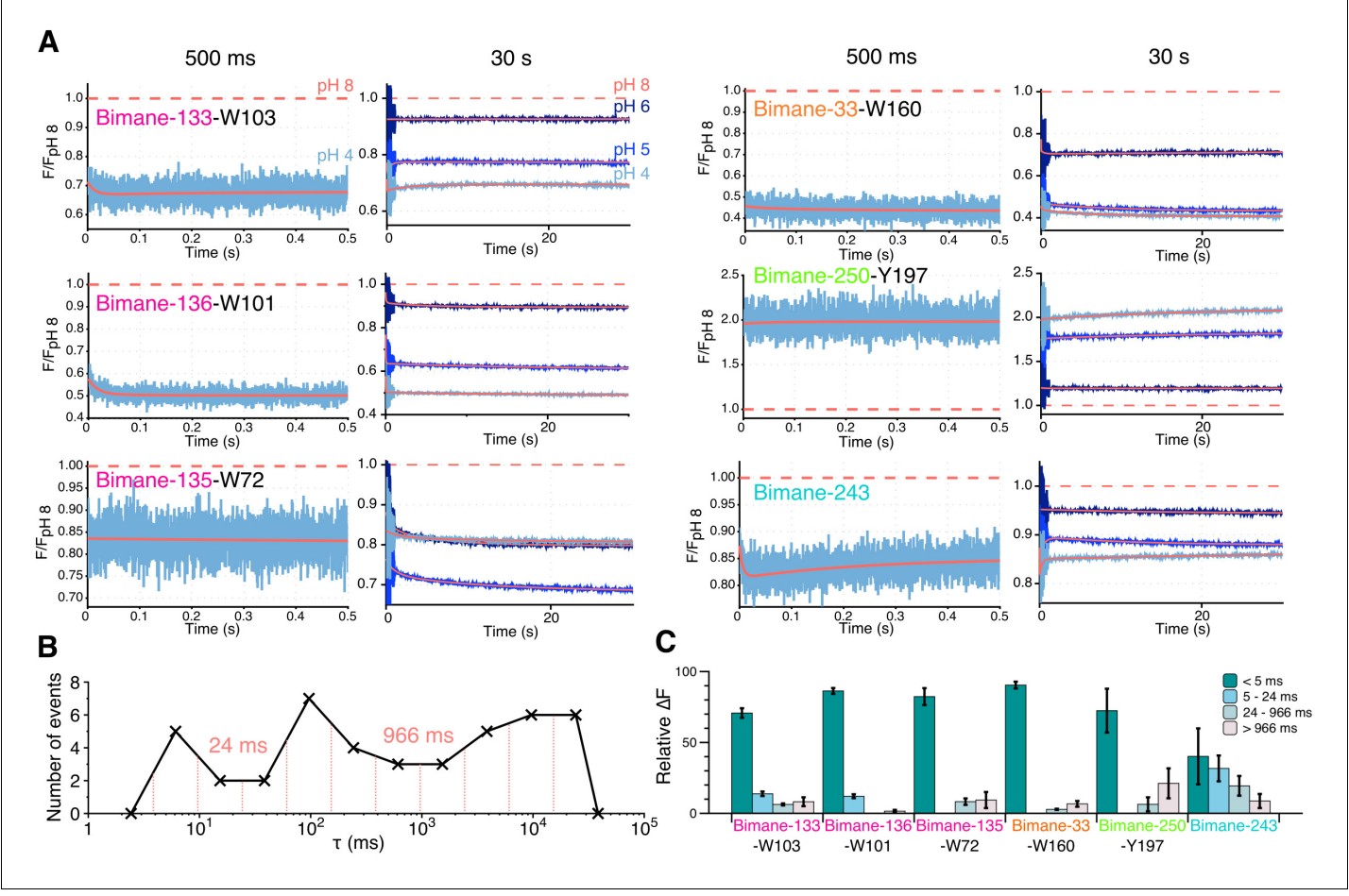

**Figure 6.** Stopped-flow fluorescence measurements. (A) Each panel shows representative traces of the major quenching mutants with a zoom on the first 500 ms (left) and the entire recording (right). All traces are fitted to a multi-exponential (see Methods) represented in red. The color code is identical in each panel with a blue gradient starting from pH 6 (dark) to pH 4 (light). The red dash line represents the starting fluorescence at pH 8 after normalization, for the raw pH 8 trace see *Figure 6—figure supplement 1*. (B) Binning of τ values extracted from the stopped-flow multi-exponential fit (all pHs and all constructs were used for the plot). The log time binning interval is of 0.4 and the resulting bins are shown by red dashed lines on the plot. The distribution shows three major clusters for which the according τ value is indicated on the plot. (C) Fluorescence variations (calculated as a % of the overall variation at 30 s) for each time interval extracted from the binning plot. In this graph, mean values are presented and error bars are calculated as standard deviations.

The following figure supplement is available for figure 6:

**Figure supplement 1.** Residuals of the stopped-flow data fits and pH 8 recordings.

measure of channel activity (see Materials and methods [*Ingólfsson and Andersen, 2010*; *Rusinova et al., 2014*]).

Both GLIC Cys-less and mutant Bimane-136-W101 show fast fluorescence decay with a rate of $\approx 80\ s^{-1}$ after incubation for 15 ms at pH 4.5 and 4.2 (*Figure 7—figure supplement 1*). This indicates robust activation of GLIC under these acidic conditions. The quenching kinetics at both pH values are identical (*Figure 7B*), suggesting that activation is already maximal at pH 4.5.

In contrast, very slow kinetics ($\approx 3\ s^{-1}$) are observed after incubating the channels for 15 ms at pH 5.2 (*Figure 7B* and *Figure 7—figure supplement 1*) which indicates that these conditions lead to little channel activation. According to these data the pH$_{50}$ for GLIC Cys-less and GLIC Bimane-136-W101 is estimated at pH 4.6 and pH 4.7, respectively (*Figure 7A*).

Increasing the pre-mix time did not increase the rates of fluorescence quenching at any of the pH values tested, suggesting that 15 ms are sufficient to reach the maximal activation of GLIC. On the

**Table 5.** Kinetic and fluorescence parameters of real-time measurements. All the values presented in the table were extracted from multi-exponential fits of the real-time measurements (see Materials and methods). The relative ΔF for each exponential fit were calculated using the maximal F variation of each curve. In the case fluorescence variations were 'bi-directional', the respective amplitudes of each phases were added to determine the maximal F variation value. (+) indicates an increase of fluorescence and (−) indicates a decrease. NA stands for non-applicable and was used when a single or double exponentials were sufficient to fit the data. For Bimane-133-W103 at pH 6, no fluorescence variations were measured after the 5 ms non-exploited data; hence no exponential fits were made for this particular condition. n represents the number of experiments. For all the data, mean values are presented and error values are calculated as standard deviations.

| Mutant | pH | Dead-time (<5 ms) Relative ΔF (%) | Exponential 1 τ1 (ms) | Relative ΔF (%) | Exponential 2 τ2 (ms) | Relative ΔF (%) | Exponential 3 τ3 (ms) | Relative ΔF (%) | n |
|---|---|---|---|---|---|---|---|---|---|
| Bimane-133-W103 | pH 6 | (−) 94 ± 6 | NA | NA | NA | NA | NA | NA | 4 |
| | pH 5 | (−) 72 ± 1 | 10 ± 4 | (−) 11.6 ± 0.7 | 290 ± 40 | (+) 11 ± 5 | 21,000 ± 6000 | (−) 5 ± 4 | 3 |
| | pH 4 | (−) 71 ± 3 | 10 ± 7 | (−) 14 ± 2 | 600 ± 400 | (+) 6 ± 1 | 26,000 ± 18,000 | (−) 8 ± 3 | 4 |
| Bimane-135-W72 | pH 6 | (−) 53 ± 9 | 50 ± 40 | (−) 12 ± 4 | 700 ± 400 | (−) 17 ± 6 | 3000 ± 1000 | (−) 19 ± 4 | 6 |
| | pH 5 | (−) 50 ± 20 | 100 ± 100 | (−) 8 ± 5 | 1000 ± 1000 | (−) 19 ± 8 | 8000 ± 4000 | (−) 21 ± 6 | 6 |
| | pH 4 | (−) 82 ± 6 | 700 ± 400 | (−) 8 ± 2 | 30,000 ± 40,000 | (−) 9 ± 6 | NA | NA | 4 |
| Bimane-136-W101 | pH 6 | (−) 40 ± 30 | 9 ± 4 | (−) 30 ± 10 | 100 ± 10 | (−) 10 ± 10 | 2900 ± 600 | (−) 14 ± 7 | 4 |
| | pH 5 | (−) 60 ± 10 | 9 ± 3 | (−) 30 ± 10 | 40 ± 20 | (−) 5 ± 2 | 18,500 ± 900 | (−) 7 ± 2 | 4 |
| | pH 4 | (−) 86 ± 2 | 11 ± 3 | (−) 12 ± 1 | 4000 ± 5000 | (−) 1.6 ± 0.9 | NA | NA | 4 |
| Bimane-33-W160 | pH 6 | (−) 73 ± 9 | 9 ± 5 | (−) 19 ± 9 | 100 ± 100 | (−) 6.1 ± 0.7 | 3000 ± 4000 | (−) 3.2 ± 0.6 | 3 |
| | pH 5 | (−) 88.3 ± 0.9 | 100 ± 100 | (−) 5 ± 2 | 1000 ± 1000 | (−) 2.9 ± 0.4 | 16,000 ± 9000 | (−) 4 ± 2 | 4 |
| | pH 4 | (−) 90 ± 2 | 100 ± 100 | (−) 2.8 ± 0.4 | 2000 ± 2000 | (−) 3.6 ± 0.8 | 10,000 ± 2000 | (−) 4.1 ± 0.5 | 3 |
| Bimane-250-Y197 | pH 6 | (+) 90 ± 10 | 300 ± 500 | (+) 10 ± 10 | NA | NA | NA | NA | 4 |
| | pH 5 | (+) 86 ± 5 | 200 ± 100 | (+) 5 ± 2 | 9000 ± 3000 | (+) 9 ± 4 | NA | NA | 4 |
| | pH 4 | (+) 70 ± 20 | 100 ± 100 | (+) 6 ± 5 | 6000 ± 5000 | (+) 20 ± 10 | NA | NA | 4 |
| Bimane-243 | pH 6 | (−) 80 ± 20 | 8000 ± 10,000 | (−) 8 ± 7 | 12,000 ± 3000 | (−) 21 ± 7 | NA | NA | 4 |
| | pH 5 | (−) 60 ± 20 | 9 ± 7 | (−) 11 ± 4 | 270 ± 40 | (+) 22 ± 8 | 16,000 ± 6000 | (−) 12 ± 7 | 4 |
| | pH 4 | (−) 40 ± 20 | 5 ± 1 | (−) 32 ± 9 | 150 ± 20 | (+) 19 ± 7 | 14,000 ± 7'000 | (+) 9 ± 5 | 4 |

contrary, prolonged incubation (above 25 ms) at low pH reduced the rates of quenching, possibly reflecting GLIC desensitization in asolectin liposomes (*Figure 7—figure supplement 1*).

For the Bimane-136-W101 mutant in asolectin liposomes, the pH-fluorescence relationship (ΔF curves), which reports on protein motions, and the pH-ion flux relationship (ΔIf curves), which reports on activity, are separated by more than one order of magnitude (*Figure 7A*). Indeed, at pH 5, the majority of the fluorescence changes are completed, whereas almost no receptors are yet active. This directly reveals an intermediate conformation of the protein, where the quaternary motion of positions 136 and 101 moving closer together has occurred, but where the channel is still closed. Interestingly, a similar scenario is seen at other positions, for which activation curves in oocytes (ΔI curves) are significantly shifted to higher proton concentrations as compared to ΔF curves, especially at positions 135 and 250 (*Figure 7C*). Since ΔI curves are shifted to the left as compared to ΔIf curves (for GLIC Cys-less and Bimane-136-W101), it is expected that ΔIf and ΔF would be even more separated for the other mutants. Hence, our data suggest that the intermediate conformation involves global motions of the protein, notably the adjacent subunits moving closer at the level of the ECD (positions 136–101 and 135–72) as well as the separation between positions 250 and 197 and thus the outward motion of the M2-M3 loop.

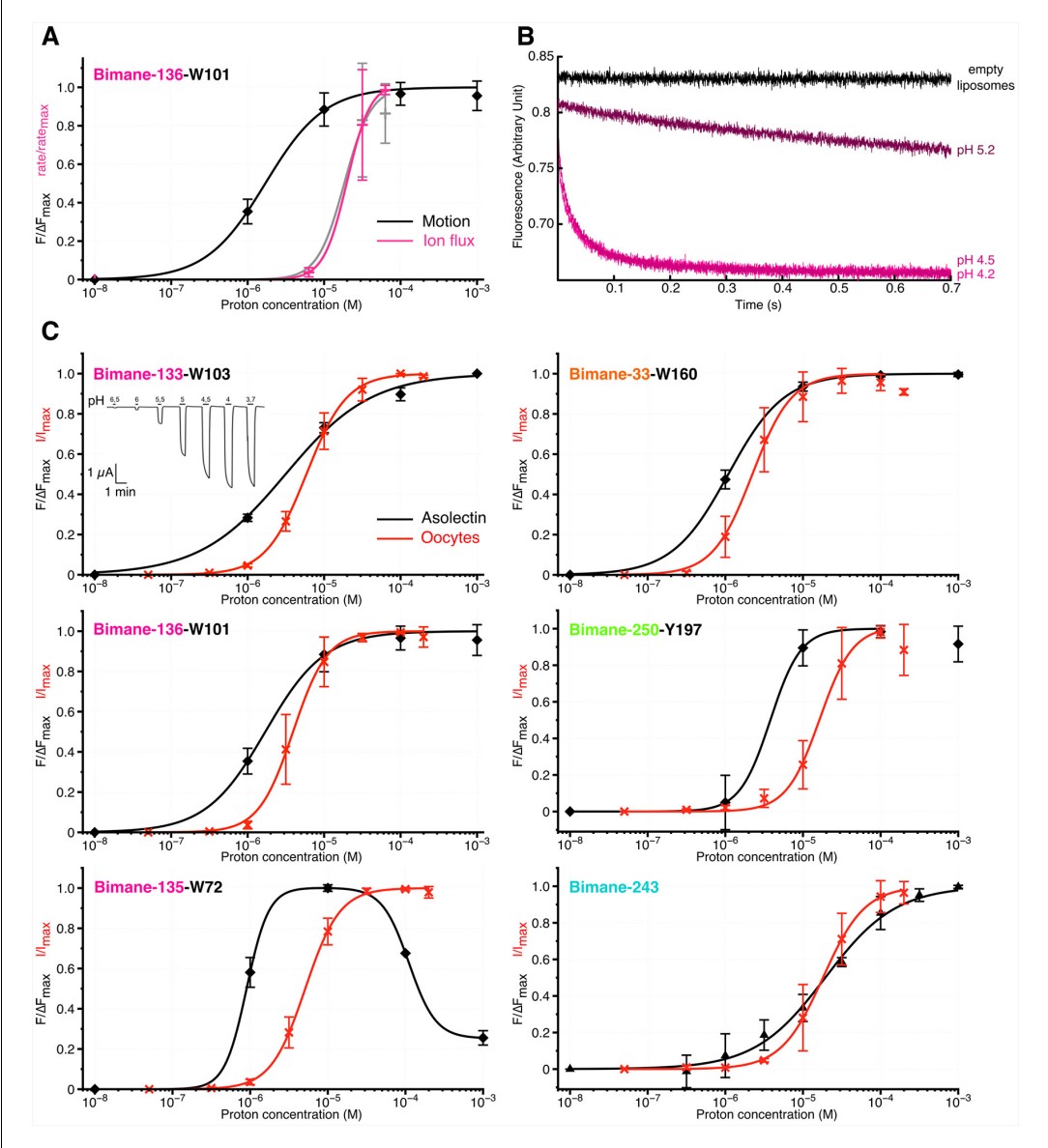

**Figure 7.** Comparison of GLIC motions and function. (A) Fits to the Hill equation of fluorescence values in asolectin liposomes (black) and rate values extracted from the thallium flux assay (pink), both normalized to their respective maximum, for the Bimane-136-W101 mutant (n = 3). For comparison the ΔIf (thallium flux assay) curve of GLIC Cys-less is shown in grey (n = 3). (B) Stopped-flow recordings of the thallium fluxes quenching assay for the Bimane-136-W101 mutant. The pre-mix time was 15 ms for the empty and pH 5.2 liposomes and of 15.5 ms for the pH 4.5 and 4.2 recordings. (C) Fits to the Hill equation for electrophysiological currents (red) and fluorescence values in asolectin liposomes (black) for all major quenching pair mutants. The current and fluorescence are normalized to their respective maximum; pH 3.7 current values were excluded from the fits for mutants Bimane-250-Y197 and Bimane-33-W160 as they were systematically smaller than the maximal current. For all the data except panel B, mean values are presented and error bars are calculated as standard deviations.

The following figure supplement is available for figure 7:

**Figure supplement 1.** Thallium flux assay.

## Discussion

We developed a series of allosteric sensors of the conformation of GLIC using the TrIQ/TyrIQ method. Upon pH drop, GLIC undergoes a cascade of allosteric transitions from the resting to the

active and desensitized states. This method allows the investigation, in a time resolved manner, of the global evolution of this mixture of states depending on the proton concentration.

While the quenching method follows the fate of mixed populations of receptors, it is informative to compare these ensemble conformational motions to those inferred from the static structures of GLIC solved by crystallography. GLIC was solved at a low proton concentration in a closed-channel conformation (pH 7) (*Sauguet et al., 2014*) and at a high proton concentration in an apparently open-channel conformation (pH 4) (*Bocquet et al., 2009*; *Hilf and Dutzler, 2009*). Interestingly, measurement of the variation of C$\beta$-C$\beta$ distances between the pH 7 and pH 4 structures parallel the motions inferred from fluorescence measurements at positions 136–101, 133–103 and 33–160 in the ECD, that are reporting the subunits moving closer together, and positions 250–197 that follow the separation of the M2-M3 loop from the top of M1 (*Supplementary file 1B*). It is thus likely that the molecular reorganizations inferred from the comparison of the two X-ray structures are contributing to the ensemble motions followed by fluorescence. However, at other positions, namely at pairs 243–238 (top of the TMD) and 136–178 (Loop C), no correlation between the structures and the population ensembles can be observed. This suggests that, at these levels, the receptors might undergo different reorganizations when GLIC is outside of the crystal, and/or that additional conformational states are contributing to the overall fluorescence signal.

A key finding of the work is the identification of an intermediate state, where the ECD compaction and the outward movement of the M2-M3 loop, monitored by bimane quenching, are not concerted with channel opening. This is directly demonstrated with the 136/101 pair, that reports nearly complete ECD compaction in steady-state conditions, at pH 5, when no ion fluxes are yet recorded (*Figure 7A*). We also show that the isomerization toward this intermediate state is very fast, since 60% of the transition is achieved in less than 2 ms at pH 5 (*Table 5*). This suggests that the isomerization toward this intermediate state precedes or is at least concomitant with activation. Indeed, previous studies identify GLIC as a slow activating channel within the pLGIC family. For GLIC reconstituted in asolectin liposomes recorded in the inside-out configuration, the activation time constant (τ) is in the 10 ms range upon activation by a very high proton concentration (pH 2.5) (*Velisetty and Chakrapani, 2012*). Likewise, in HEK cells, out-side-out patch clamp recordings under fast perfusion, at pH 4, reveal activation time constants ranging from 30 to 150 ms (*Laha et al., 2013*). While the activation kinetics of GLIC in liposomes could not be resolved in the present study, it is likely that the very fast transition toward the intermediate state is part of the activation mechanism. Altogether, we propose a kinetic scheme where a global 'pre-activation' step first occurs in under 2 ms, involving the whole ECD compaction and the outward motion of the M2-M3 loop, followed by a slower and more localized transition for channel opening (*Figure 8*). This latter motion may plausibly be related to the fluorescence changes observed in the 5–966 ms ranges, that are particularly marked for the 243 reporter which is located near the channel gate.

Several mutants and a wild-type GLIC C-terminally tagged with 10 histidines, were found to crystallize in different locally-closed 'LC' conformations characterized by an overall conformation of the ECD nearly identical to GLIC$_{pH\ 4}$, but with a closed channel (*Prevost et al., 2012*; *Gonzalez-Gutierrez et al., 2013*). Among these, the structures of GLIC bearing a disulphide bridge C33-C245, as well as the single mutant GLIC E243P, show a particularity with the M2-M3 loop revolved outwards, identically as for GLIC$_{pH\ 4}$, but still with a closed ion channel. Hence, we may speculate that this particular LC conformation could be a candidate for the pre-active state of GLIC, since it shows compaction of the ECD and motion of the M2-M3 loop without channel opening.

We also observe molecular reorganizations occurring on longer time scale. First, quenching experiments show small fluorescence changes in the 966 ms–30 s range for all positions. Second, ion flux experiments show a decrease in channel activity when GLIC is pre-incubated with protons for a more than 100 ms. These events could thus be linked to receptor desensitization. Indeed, in HEK cells, out-side-out patch clamp recordings show desensitization to be biphasic, with a fast component observed on half of the patches (τ around 200 ms), followed by a slow component observed in all patches (τ around 10 s) (*Laha et al., 2013*). Interestingly, previous analysis of the desensitization mechanism of GLIC by EPR (*Velisetty and Chakrapani, 2012*), and of other pLGICs by mutagenesis (*Gielen et al., 2015*) and NMR (*Kinde et al., 2015*), support the view that desensitization proceeds through a narrowing of the lower part of the channel. Additionally, recently published X-ray structures of a GABA$_A$ receptor (*Miller et al., 2014*) and of the α4β2 nAChR (*Morales-Perez et al., 2016*) were proposed to correspond to a desensitized conformation and show a channel constriction

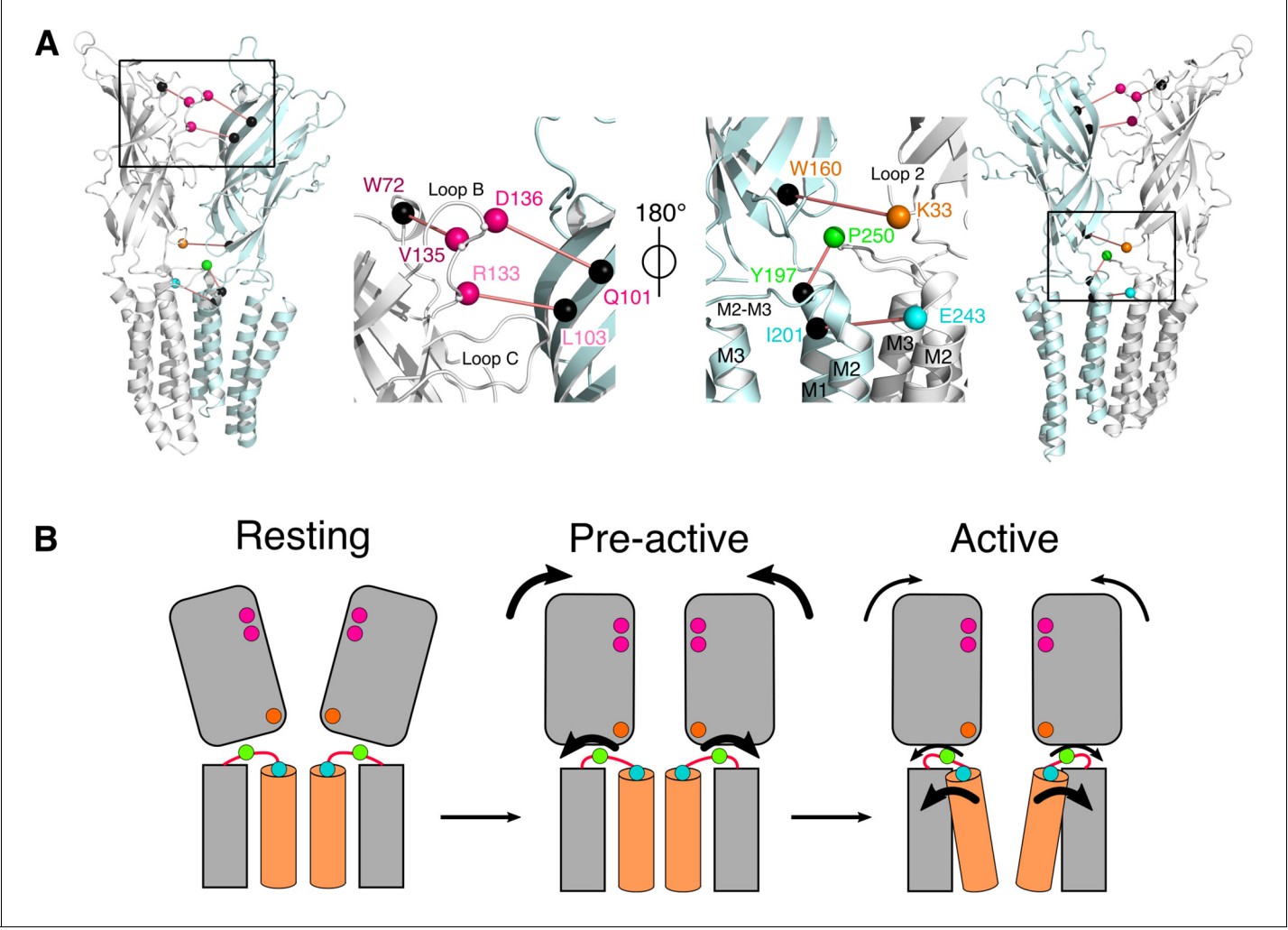

**Figure 8.** Conformational motions summary. (A) Global and zoomed in view of the GLIC$_{pH\ 7}$ structure, representing only two subunits, with the left structures viewed from the outside of the pentamer and the right structures from the inside. All major quenching pairs are represented by a sphere, colored for the mutation to cysteine and bimane labeling, and black for the corresponding quencher. All pairs are captioned in the same color and joined by a red line. (B) Hypothetic kinetic scheme for the GLIC structural reorganizations. The large arrows represent the major recorded fluorescence variations and the small arrows represent minor variations.

in the bottom part of the ECD. Therefore, desensitization might occur through reorganizations in the TMD, with comparatively smaller motion at the ECD and particularly at the ECD-TMD interface.

In conclusion, the monitoring of electrophysiologically silent states by the TrIQ/TyrIQ method allows us to structurally describe an intermediate state and propose a pre-activation mechanism. Interestingly, available structures of eukaryotic pLGICs, namely the GluClαR (*Hibbs and Gouaux, 2011*; *Althoff et al., 2014*) and the α1GlyR (*Du et al., 2015*), also suggest a quaternary compaction of the ECD upon agonist binding, as well as outward motions of the M2-M3 loop upon channel opening that are similar to that observed on GLIC. In addition, φ-value analysis following extensive mutational analysis of the muscle nAChR further suggest that the top of the ECD around the orthosteric pocket, as well as the M2-M3 loop move early in the course of the gating transition toward the active state (*Purohit et al., 2013*). The present study thus not only sheds light on the gating mechanism of the pLGIC model GLIC, but also provides a structural template to investigate the gating of eukaryotic and mammalian receptors.

## Materials and methods

### Buffers and chemicals

Buffer A consists of 20 mM Tris and 300 mM NaCl, adjusted to pH 7.4 unless otherwise stated.

Buffer B consists of 300 mM NaCl, 2.7 mM KCl, 5.3 mM $Na_2HPO_4$ and 1.5 mM $KH_2PO_4$, adjusted to pH 8 unless otherwise stated. Both buffers were supplemented with 0.02% DDM (Anatrace, Maumee, OH) when specified.

Buffer C consists of 15 mM $Na_2HPO_4$, 150 mM $NaNO_3$ adjusted to pH 7.

Pre-mix buffer consists of 10 mM $Na_2HPO_4$, 140 mM $NaNO_3$ adjusted to pH 7 unless otherwise stated.

Quenching buffer consists of 10 mM Na2HPO4, 90 mM TlNO3, 50 mM NaNO3 adjusted to pH 7 unless otherwise stated.

MBS buffer consists of 88 mM NaCl, 1 mM KCl, 2.5 mM $NaHCO_3$, 5 mM HEPES, 0.7 mM $CaCl_2$ and 1 mM $MgSO_4$.

MES buffer consists of 100 mM NaCl, 3 mM KCl, 1 mM $CaCl_2$, 1 mM $MgCl_2$ and 10 mM MES.

CHO labeling buffer consists of 150 mM NaCl, 8.1 mM $Na_2HPO_4$, 1.9 mM $NaH_2PO_4$, 0.1 mM $CaCl_2$ and 1 mM $MgCl_2$, adjusted to pH 7.4.

CHO conservation buffer consists of 160 mM NaCl, 4.5 mM KCl, 2 mM $CaCl_2$, 1 mM $MgCl_2$, 10 mM HEPES and 8 mM glucose, adjusted to pH 7.4.

Monobromo bimane (ThermoFisher Scientific, Pittsburg, PA) was dissolved in 100% DMSO at 10 mM and stored at −20° C.

Bimane Bunte salt was dissolved at 50 mM in water and stored at −80° C.

Unless otherwise stated, all chemicals were purchased from Sigma Aldrich (St Louis, MO).

### Mutagenesis

All GLIC constructs were generated using the molecular probe mutagenesis kit (ThermoFisher Scientific) on the Cys-less background (C27S) in two vectors previously described (*Bocquet et al., 2007*), constructed as follows:

- The pmt3 vector for eukaryotic expression with an α7 peptide signal and an HA-tag in the C-terminal of GLIC.
- The pet20b vector for prokaryotic cells, expressing GLIC as an MBP (in the N-terminal) fusion protein.

### Protein production and fluorophore labeling

MBP-GLIC was produced as previously described (*Bocquet et al., 2009*), in BL21 *E.Coli* strains. Briefly, MBP-GLIC was produced at 20° C in BL21 bacteria after IPTG (20 mM) induction. After cell disruption by sonication, membranes were separated through ultracentrifugation (40,000 rpm) and MBP-GLIC extracted in buffer A 2% DDM overnight. MBP-GLIC was purified on an amylose resin and eluted by maltose addition in buffer A 0.02% DDM. Proteins were run on a gel filtration (superpose 6 10/300 GL (GE Healthcare, Chicago, IL)) column for removal of remaining maltoporin contaminants. The MBP was subsequently cleaved from GLIC by addition of thrombin (Merck Millipore, Billerica, MA) overnight, and GLIC purified again through gel filtration in buffer A 0.02% DDM. For receptors with the K33C and E243C mutations, the protein was incubated for 1 hr with 10 mM DTT prior to the gel filtration to reduce potential disulfide bridges. Pentameric GLIC was then incubated with mBBr at a five molar excess ratio (monomer 1:5 mBBr) while ensuring the final DMSO concentration did not exceed 1%, overnight, under agitation, at 4°C. Excess fluorophore was removed by gel filtration and GLIC-labeled samples flash frozen for storage at −80°C.

### Liposome reconstitution

Powdered asolectin extracted from soybean (Sigma Aldrich) was solubilized in buffer B at a concentration of 10 mg/mL using a potter, and either used fresh or aliquoted and frozen at −20°C for later uses. GLIC was reconstituted in liposomes with a 1:5 GLIC/asolectin w:w ratio. For 200 µg of protein reconstitution, the following procedure was used and all the steps performed at room temperature: 1 mg of asolectin in buffer B was mixed with DDM to reach a final DDM concentration of 0.7% and the solution equilibrated for 40 min to solubilize the pre-formed multilayered liposomes. The 200 µg

of protein in buffer A 0.02% DDM was added to the solution and the overall volume brought to 1 mL to lower the DDM concentration to 0.2%, followed by equilibration for 1.5 hr under gentle agitation. For liposome formation and gentle inclusion of the protein, detergent was removed by incremented step addition of SM2 Bio-Beads (50 mg, 150 mg and 300 mg (Bio-Rad, Hercules, CA)) pre-activated using methanol. Bio-beads were removed through light centrifugation and the proteoliposomes were either used fresh or stored at 4°C for a maximum of 4 days.

## Steady-state fluorescence recordings

All fluorescence recordings were done on a Jasco 8200 fluorometer (MD, USA). Buffers and proteins were equilibrated at room temperature before each recording which were made at 20°C. mBBr and Bimane-GLIC samples were excited at 385 nm and their emission spectra were recorded from 420 to 530 nm through 2.5 nm slits at excitation and emission. Scan speed, sampling and PMT values were kept constant for all measures for subsequent comparisons. The membrane receptor GLIC being activated by protons ($pH_{50}$ = 5.3)(*Bocquet et al., 2007*), we confirmed that both the mBBr fluorescence and its quenching properties are identical in proton concentrations ranging from pH 9 to pH 2 (*Figure 3—figure supplement 1A,B*). Control experiment with the Cys-less GLIC treated with mBBr show negligible non-specific fluorescence (*Figure 3—figure supplement 1C*). All measurements were made on 1 mL of protein sample in disposable UV transparent 2.5 mL cuvettes (Sigma). As a consequence to the small volumes used, it was not possible to precisely acidify the protein sample using concentrated acid, or diluted acid without considerably changing the final concentrations of ions and protein. Hence, measurements were made after mixing one volume of protein in buffer A 0.02% DDM, with one volume of buffer B 0.02% DDM previously acidified by 1 M HCl to reach, after mixing, the desired pH value. The time necessary for the sample mixing and recording of fluorescence was evaluated to be approximately 30 s; this time is sufficient to reach steady-state conditions as shown by the plateau of fluorescence variations recorded on the stopped-flow after 20 s (*Figure 6A*). The same mixing protocol was used for the protein samples reconstituted in asolectin. For every recording, the fluorescence emission spectra were stable over multiple recordings, not showing signs of fluorophore bleaching. For each pH tested, tryptophan emission spectra were recorded (excitation at 280 nm) to ensure the proteins did not suffer denaturation and all fluorescence changes were reversible upon return to pH 7 (*Figure 3—figure supplement 2*). For the propofol recordings, the bimane-labeled mutants were recorded, followed by addition of propofol to a final concentration of 100 μM, and re-recording of the sample. In these experiments, the tryptophan fluorescence was not followed as the propofol produces a strong contamination signal at these excitation and emission wavelengths.

## Stopped-flow fluorescence recordings

Recordings were made on a SFM-300 stopped-flow apparatus (Bio-Logic, Seyssinet-Pariset, France). To ensure lamp and temperature stability, the apparatus was turned on and equilibrated at 20°C by temperature-controlled circulating water for at least 1 hr before recordings. Excitation was set at 385 nm through an 8 nm slit and emission fluorescence recorded through a 420 nm high-pass filter. A two-syringe injection system was used where syringe one was loaded with the protein sample in buffer B at pH 8, and syringe two loaded with buffer B equilibrated at specified pHs. The injection volume was 150 μL for each syringe at an injection speed of 8.7 mL/s (total speed of 17.4 mL/s). The filling of the FC15 recording flow cell (0.15 × 0.15 cm, 35 μL total) was thus achieved after a theoretical 2.1 ms mixing dead-time (manufacturer's information). To follow with good time resolution the first fast events, 30 s long recordings were made through three sampling times: the first 500 ms were recorded with a 100 μs sampling time, followed by 0.5 to 1.5 s with a 1 ms sampling time, and 1.5 to 30 s with a 50 ms sampling time. The first 3 ms of recordings were excluded from the analysis because of non-reproducibility, yielding a 5.1 ms total 'non-analyzed' data (3 ms + 2.1 ms dead-time). For each condition, a mean of 10 recordings was counted as n = 1 and all conditions were recorded with n = 3 or 4. Each data set at pH 6, pH 5 and pH 4 was normalized to the integral 30 s recording made at pH 8 on the same day and same batch of proteoliposomes, to correct potential unspecific drifting/bleaching of fluorescence during the recordings. Each mean of 10 was analyzed individually using Datagraph (Visual data tools) and fitted over the total 30 s of recording to a maximum three exponentials equation:

$$y(x) = F + F1 * e^{-k1*x} + F2 * e^{-k2*x} + F3 * e^{-k3*x}$$

where F represents the final fluorescence intensity, F1,2,3 represent the Δfluorescence of a particular exponential phase and k1,2,3 the kinetic constant for each exponential phase in $s^{-1}$. Systematic background fluorescence recordings of buffers were done and subtracted from the data. For residuals of the fits, see *Figure 6—figure supplement 1*.

## Thallium fluxes assay

A sequential-mixing stopped flow spectrofluorimeter (SX.20, Applied Photophysics) was employed to assay GLIC channel activity by measuring the $Tl^+$-induced fluorescence quenching of a liposome-encapsulated ANTS (8-Aminonaphthalene-1,3,6-Trisulfonic Acid, Disodium Salt, Life Technologies, NY) fluorophore via $Tl^+$ influx through channels as previously described (*Rusinova et al., 2014*; *McCoy et al., 2014*; *Posson et al., 2015*).

### Proteoliposomes preparation

Asolectin lipids (Sigma-Aldrich) were dissolved in chloroform, dried to a thin layer under a constant $N_2$ flow in round bottom flasks and further dried over night under vacuum. Lipids were rehydrated to a concentration of 10 mg/ml in buffer C with 33 mM CHAPS by sonication. Water-solubilized ANTS was added to a final concentration of 25 mM followed by GLIC Cys-less or Bimane-136-W101 to a concentration of 30 µg/mg lipid. Detergent was removed by incubating the mix in the presence of 25% SM-2 BioBeads (Bio-Rad) for 2 hr under constant agitation followed by a 20 s sonication in a bath sonicator. After extrusion (Mini extruder, Avanti Polar Lipids, 0.1 µm membrane (AL, USA)) the vesicles have a mean diameter of 150 nm (*Ingólfsson and Andersen, 2010*). Extra liposomal ANTS was removed via a PD-10 desalting column (GE Healthcare) and the buffer was exchanged to the pre-mix buffer. For the recordings, the proteoliposome solutions were diluted fourfold to ensure a good signal-to-noise ratio and allow multiple experiments using the same reconstitution.

### Stopped-flow recordings

Recordings were made using a SX20 stopped-flow (Applied Photophysics, Surrey, UK). Proteoliposomes and pre-mix buffer, pre-equilibrated to reach final pH values of 7 to 3.6 after mixing, were injected with a 1:1 dilution in the aging loop. After delay times ranging from 10 to 200 ms, the solution of the aging loop was mixed with the quenching buffer in a second injection step (1:1 dilution) still maintaining the pH value reached after the first mixing. Taking in account the machine deadtime, the shortest experimental delay time reached was of 15 ms. Reference measurements were performed in the absence of $Tl^+$ using pre-mix buffer for the second injection. ANTS fluorescence was recorded for 1 s through a 455 nm high-pass filter after excitation at 352 nm. Picrotoxin block experiments were performed as described above by supplementing the pre-mix buffer with 800 µM and the quenching buffer with 400 µM picrotoxinin, respectively. Identical control experiments were performed using protein-free liposomes. Three independent reconstitutions into liposomes were analysed for each GLIC variant and at least seven repeats were recorded for every condition tested. The day-to-day variations of quenching rates obtained from different reconstitutions were smaller than 10%.

Fluorescence traces (first 100 ms) were fit with a stretched exponential as previously described (*Ingólfsson and Andersen, 2010*; *Rusinova et al., 2014*) (*Equation 1*) in Matlab (MathWorks, Natick, MA) (*Rusinova et al., 2014*) and the quenching rates at 2 ms were calculated (Equation 2) (*Berberan-Santos et al., 2005*) and averaged over all repeats.

$$F(t) = F_\infty + (F_0 - F_\infty) \cdot e^{-\left(\frac{t}{\tau_0}\right)^\beta} \tag{1}$$

$$k(2\,ms) = \left(\frac{\beta}{\tau_0}\right) \cdot \left(\frac{2\,ms}{\tau_0}\right)^{(\beta-1)} \tag{2}$$

with $F_\infty$ and $F_0$ denoting for the final and the initial fluorescence values, respectively, $\tau_0$ the time constant, $\beta$ the exponential-stretch parameter, and $k$ (2 ms) the rate constant at 2 ms.

## Electrophysiological recordings

Functional recordings of GLIC were made on *Xenopus* oocytes provided by the Centre de Ressources Biologiques Xénopes–Rennes (France). Electrophysiological recordings were made as previously described (*Duret et al., 2011*) after 48–96 hr of expression, with the difference that oocytes were clamped at −40 mV for recordings. For all mutants, with the exception of P250C-Y197F (see below), currents were recorded on non-labeled oocytes, followed by BBs labeling and re-recording on the same oocytes. Independent mutants-expressing oocytes were also used for current recordings after labeling without prior non-labeled recordings. Both methods led to similar results and the $pH_{50}$ and nH values are given for a mix of oocytes recorded with the different methods. The P250C-Y197F mutant showed a notable run-down after activation preventing several consecutive recordings of the same cell. In this case, currents after BBs labeling were recorded using the second method only. BBs labeling of oocytes was obtained after a 1 hr incubation at room temperature, under very gentle agitation, in MBS 1 mM BBs. Oocytes expressing mutants K33C or E243C were treated with 10 mM DTT 10 min prior to BBs labeling. After labeling, all oocytes were rinsed in MBS buffer and recorded within 1 hr post-labeling. Mutants leading to currents smaller than 500 nA at high proton concentrations (pH 4) were categorized as non-functional. For all non-functional mutants, expression tests were performed through immunolabeling of oocytes (*Figure 2—figure supplement 5*). Electrophysiological recordings were analyzed using AxoGraph X and ClampFit (Molecular Devices, Sunnyvale, CA). The Hill equation was used for the dose-response fits:

$$y(x) = \frac{a * x^{nH}}{x^{nH} + EC_{50}^{nH}}$$

where a represents the maximal current value after normalization, nH represents the hill number and $EC_{50}$ the proton concentration for which half of the maximal electrophysiological response is recorded. A double exponential equation was used to fit the activation currents of GLIC labeled with bimane:

$$y = A1 * \left(1 - e^{-\frac{t}{\tau 1}}\right) + A2 * \left(1 - e^{-\frac{t}{\tau 2}}\right) + C$$

where t is time, A1/A2 represent the current amplitude of the fast and slow activation phases respectively, C accounts for the current value at the end of the fit and τ1/τ2 represent the activation kinetics. A weighted activation constant was also calculated using the following equation:

$$\tau w = \left(\left(\frac{A1}{A1 + A2}\right) * \tau 1\right) + \left(\left(\frac{A2}{A1 + A2}\right) * \tau 2\right)$$

## *Xenopus* oocytes immunolabeling

Immunolabeling on *Xenopus* oocytes expressing GLIC WT or mutants was performed as previously described (*Sauguet et al., 2014*). Briefly, oocytes were co-injected in the nucleus with a mix of two separate pmt3 vectors containing either the cDNA of GLIC-HA (80 ng/µl) or GFP (10 ng/µl). Control oocytes were injected with the GFP alone. After 72 hr of protein expression, GFP positive cells were fixed in 4% paraformaldehyde (4°C, O/N), blocked in PBS + 4% horse serum (30 min, RT [Sigma Aldrich]), and immunolabeled in PBS + 2% horse serum using a rabbit anti-HA primary antibody (1.5 hr) and an anti-rabbit Cy5 coupled secondary antibody (1 hr, RT ([ThermoFisher Scientific]). Oocytes were then re-fixed with 4% paraformaldehyde (4°C, O/N), placed into 3% low-melting agarose blocks, and subsequently sliced at 40 µm intervals. Slices of three different oocytes per constructs were analyzed using epi-fluorescence microscopy with constant exposure times.

## CHO culture, transfection and labeling

Chinese hamster ovary (CHO-K1 CCL-16 from ATCC, USA) cells were cultured in Ham's F-12K Kaighn's modification medium (ThermoFisher Scientific) supplemented with 10% fetal calf serum and containing penicillin (100 U/mL) and streptomycin (100 µg/mL). As the CHO cells were solely used for the bimane membrane-permeability test, they were not tested for mycoplasma contamination. Cells were plated in 35 mm diameter polystyrene plates (Corning, Corning, NY) and transfected with GLIC cDNA and mCherry cDNA, as a transfection control, using a JetPRIME kit (Polyplus Transfection, Illkirch, France) according to the manufacturer's instructions. Protein expression was allowed

for 48–72 hr before cell labeling. Throughout the labeling process, cells and buffers were kept at 4°C to avoid endocytosis of the fluorophore. For labeling with BBs, cells were rinsed three times (5–10 min, gentle agitation, 4°C) in labeling buffer and incubated for 1 hr with 1 mM BBs in labeling buffer. They were then rinsed two times with labeling buffer (5–10 min, gentle agitation, 4°C) and kept in conservation buffer at room temperature during the imaging process. Each observation was made at least three times on different cell batches.

## Confocal microscopy

Confocal laser scanning of fluorescence was performed using an Ultima scanning head (Bruker Fluorescence Microscopy, Middleton, USA) mounted on an Olympus BX61W1 microscope and equipped with a 60x (1.1 NA, Olympus Optical, Tokyo, Japan) water immersion objective. Bimane and mCherry were excited at 405 nm with a laser power of 1.5–2.5 μW and at 561 nm with a laser power of 0.8–4 μW, respectively. Emitted fluorescence was collected through the same objective lens, and focused on a 150 or 100 μm pinhole ($\approx$ 1 to 1.5 Airy unit), placed on a conjugate image plane (the confocal pinhole). Fluorescent emission from Bimane was filtered with a 525/50 nm band pass filter and detected in gallium arsenide phosphide-based photocathode photomultiplier tube (H7422P, Hamamatsu Photonics, Hamamatsu, Japan). mCherry fluorescence was filtered with a 605/70 band pass filter (all filters were from Chroma, Taoyuan City, Taiwan) and detected in a side-on multi-alkali PMT (3896, Hamamatsu Photonics).

## Crystallography

The bimane labeled GLIC V135C was crystallized in the same conditions as WT GLIC (*Sauguet et al., 2013*). The crystals were directly flash-frozen in liquid nitrogen prior to data collection. Data sets were collected on the PROXIMA1 beamline of the SOLEIL synchrotron, Gif-sur-Yvette, France. Reflections were integrated using XDS (*Kabsch, 2010*) and further processed using programs from the CCP4 suite (*Winn et al., 2011*). As expected, the crystals were isomorphous to the previously described crystal lattice of the open receptor and belonged to space group C121 (unit-cell parameters: a = 182.034 Å, b = 134.075 Å, c = 159.945 Å, $\alpha = \gamma = 90.00°$, $\beta = 102.51°$) with one pentamer in the asymmetric unit (see *Supplementary file 1A*).

The phases were directly calculated by performing rigid-body refinement with REFMAC5 (*Murshudov et al., 2011*) using PDB entry 3EAM (*Bocquet et al., 2009*) as a starting model. The structure was then subjected to restrained refinement with REFMAC5 using NCS restraints. As the covalent link between the bimane and the cysteine side chain is non-standard, its description was defined in an additional library and incorporated in the pdb file. The resulting model was subsequently refined by BUSTER (*Blanc et al., 2004*). The final structure was validated using the MolProbity web server (*Chen et al., 2010*).

The PDB accession code is 5IUX.

## Bimane bunte salt synthesis

### General

Commercially available reagents were used throughout without further purification. Analytical thin layer chromatography was performed on Merck 60 F-254 precoated silica (0.2 mm) on glass and was revealed by UV light. $^1$H and $^{13}$C NMR spectra were recorded on a Bruker AC 300 apparatus at 300 MHz and 75 MHz, respectively. The chemical shifts for $^1$H NMR were given in ppm downfield from tetramethylsilane (TMS) with the solvent resonance as the internal standard. HRMS (ESI) analysis was performed with a time-of-flight mass spectrometer yielding ion masse/charge (*m/z*) ratios in atomic mass units. Purity of synthesized compound was determined by reverse phase HPLC using a 150 mm x 2.1 mm (3.5 μm) C18-column: the compound was eluted over 24 min with a gradient from 95% ACN/5% ($H_2O$ + 0.1% $HCO_2H$) to 5% ACN/95% ($H_2O$ + 0.1% $HCO_2H$). Note that the use of formic acid in elution solvent gave the corresponding thiosulfonic acid.

### Typical procedure for bimane bunte salt (BBs) (*Reeves et al., 2014*)

To a solution of bromobimane (50 mg, 0.18 mmol, 1.0 equiv.) in a mixture of water (105 μL) and MeOH (300 μL), was added sodium thiosulfate pentahydrate (54.6 mg, 0.22 mmol, 1.2 equiv.). The reaction was stirred at 65°C for 5 hr and cooled to room temperature. After evaporation of the

solvent under reduced pressure, the crude reaction product was treated with MeOH (0.5 mL), heated to 50°C and filtered. The filtrate was concentrated in vacuo to furnish a yellow solid which was tritured with hexane, filtered and dried at 50°C under vacuum providing the expected BBs in 80% yield (47 mg) as a yellowish solid. The 81.4% purity of BBs ($t_R$ = 11.65 min) was determined by reverse phase HPLC ($\lambda$ = 235 nm). This solid did not display a melting point but rather decomposition. $^1$H NMR (300 MHz, $D_2O$): $\delta$ 4.43 (s, 2 hr), 2.47 (s, 3 hr), 1.93 (s, 3 hr), 1.83 (s, 3 hr); $^{13}$C NMR (75 MHz, $D_2O$/MeOD): $\delta$ 164.0, 163.4, 150.8, 148.2, 115.3, 112.7, 12.1, 7.1, 6.6; HRMS (*m/z*): [M + H]$^+$ calculated for $C_{10}H_{13}N_2O_5S_2$, 305.0260; found 305.0260.

## Acknowledgements

We would like to thank Alain Chaffotte for his help in setting up the bimane stopped-flow experiments and DJ Posson for assistance with initial double mixing stopped-flow experiments. We thank OS Andersen and R Rusinova for providing us with time at the SX.20 stopped-flow instrument, which is maintained by the NIH grant R01GM021342 to OS Andersen. We thank Nelson Rebola and David DiGregorio for assistance in the confocal microscopy recordings. We also thank Karima Medjebeur for aid in protein production and Jean-Pierre Changeux, Marc Gielen and Akos Nemecz for critical reading of the manuscript. We acknowledge financial support by the Agence Nationale de la Recherche 'ANR pentagate' (AM) and the Foundation de la Recherche Médicale 'Equipe FRM DEQ20140329497'.

## Additional information

### Funding

| Funder | Grant reference number | Author |
|---|---|---|
| Agence Nationale de la Recherche | Pentagate | Anaïs Menny<br>Solène N Lefebvre<br>Emmanuelle Drège<br>Zaineb Fourati<br>Marc Delarue<br>Delphine Joseph<br>Pierre-Jean Corringer |
| Centre National de la Recherche Scientifique | UMR 3571 | Anaïs Menny<br>Solène N Lefebvre<br>Pierre-Jean Corringer |
| Institut Pasteur | | Anaïs Menny<br>Solène N Lefebvre<br>Zaineb Fourati<br>Marc Delarue<br>Pierre-Jean Corringer |
| Université Pierre et Marie Curie | PhD student fellowship | Anaïs Menny<br>Solène N Lefebvre |
| Fondation pour la Recherche Médicale | DEQ20140329497 | Anaïs Menny<br>Pierre-Jean Corringer |
| Centre National de la Recherche Scientifique | UMR 8076 | Emmanuelle Drège<br>Delphine Joseph |
| Centre National de la Recherche Scientifique | UMR 3528 | Zaineb Fourati<br>Marc Delarue |
| National Institutes of Health | R01 GM088352 | Crina M Nimigean |

The funders had no role in study design, data collection and interpretation, or the decision to submit the work for publication.

### Author contributions

AM, Designed most experiments, Carried out all aspects of mutagenesis, biochemistry, bimane fluorescence and electrophysiology experiments, Analyzed all the data, Wrote the manuscript and its revision; SNL, Assisted with liposome experiments and data analysis, Participated in the revision of

the manuscript; PAMS, Performed thallium flux experiments and data analysis, Assisted with preparing the manuscript; ED, Designed and Synthetized the bimane Bunte salt; ZF, Performed crystallographic data collection, Crystallographic data analysis; MD, Supervised crystallographic experiments, Data validation; SJE, Assisted in stopped-flow data analysis, Assisted with preparing the manuscript; CMN, Designed thallium flux experiments and supervised data analysis, Assisted with preparing the manuscript and its revision; DJ, Designed the bimane bunte salt, Assisted with preparing the manuscript and its revision; P-JC, Designed most experiments, Supervised the project, Wrote the manuscript and its revision

### Author ORCIDs
Anaïs Menny, http://orcid.org/0000-0002-6044-4119
Pierre-Jean Corringer, http://orcid.org/0000-0002-4770-430X

## Additional files

### Supplementary files
• Supplementary file 1. The supplementary file contains two tables presenting the X-ray diffraction data and analysis for the mutant Bimane-135 (*Supplementary file 1A*) and the distance analysis of the GLIC wild-type structures (*Supplementary file 1B*).

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
