## [Decision Letter]

Thank you for submitting your article "Identification of a pre-active conformation of a pentameric channel receptor" for consideration by *eLife*. Your article has been favorably evaluated by Richard Aldrich (Senior Editor) and three reviewers, one of whom, Baron Chanda (Reviewer #1), is a member of our Board of Reviewing Editors. The following individuals involved in review of your submission have agreed to reveal their identity: Lucia Sivilotti (Reviewer #2); Ryan Hibbs (Reviewer #3).

The reviewers have discussed the reviews with one another and the Reviewing Editor has drafted this decision to help you prepare a revised submission.

Summary:

The proton gated GLIC channels have become excellent model systems to understand how structural changes determine the function of pentameric ligand-gated ion channels. Despite the fact that there are now multiple high-resolution structures of GLIC channels and its mutants, our understanding of how proton activates these channels remain limited. This is in large part due to absence of experimental studies to probe dynamics of GLIC channels. Current models of ligand activation are based on studies of other pentameric ion channels which, on the other hand, are not as well characterized from a structural standpoint. In this study, Menny et al., map the time-resolved structural changes in GLIC using fluorescent probes and stopped flow kinetics. The authors find that the GLIC channels transition into a pre-active conformation within the first few milliseconds when activated by pH jump and that these transitions in the ligand binding domain precedes channel opening. This clearly is an important contribution to the field and provides new insight into the nature of the flipped state. Nevertheless, the reviewers have few key concerns that should be addressed in the revised version.

Essential revisions:

1) It is not clear whether the authors have conclusively ruled out the possibility that these effects on fluorescence quenching is not due to titration of di-amide group in bimane. I am not sure what the pKa of this group is but pKa of amide carbonyl is pH 3.0 and it is important to rule out the possibility that these changes in fluorescence are not due to protonation/deprotonation. For instance, one could argue that the differences in pH sensitivity just reflects the changes in pKa of the di-amide carbonyls due to the local environment rather than a conformational change. Although, the authors have shown that the fluorescence is unaffected going from pH 4.0 to pH 7.0, they cannot conclusively rule out the alternate possibility. This can be addressed by discussing these caveats when interpreting their data. In addition, the authors may want to show that the dye fluorescence is unaffected even the pH titration is extended by at least one pH unit on both ends.

2) It is nice to see that the 243 position shows additional slow kinetics that is in the same range as channel opening but actual change compared to the large baseline noise is pretty small. Can the authors rule out alternate possibility that this small decrease is not due to bleaching, for instance? The fact that the pH titration curves at 243 position for TRIQ and Tl+ flux overlaps, is a strong evidence that this position probably reports on channel opening.

3) Introduction, last paragraph: quenches the fluorophore when the distance is less than… Could you expand on that? Once you get below 15 Å for Trp, is there a relation between the effectiveness of quenching and the distance? For instance – D136 is always quenched by W93. This suggests that their distance is less than 15 Å in all conditions, but does it exclude this distance changing during activation, while remaining below 15?

4) Steady state quenching measurements at pH values < 5 are likely reporting on a desensitized conformation, correct? The results in the steady-state sections are framed largely from a context of understanding activation, but at low pH, from studies on patches, I would expect the receptor to be desensitized. Clarification of this point in a rebuttal and/or in the text (e.g. see subsection “The six fluorescent sensors report allosteric reorganizations of GLIC”, first paragraph) would be helpful to me.

5) Presumably, in liposome reconstitution experiments beginning at the subsection “Detergent-solubilized and lipid-reconstituted GLIC show similar pH-elicited reorganizations”, ~50% of the receptors have their ECDs (and H^+^ binding sites) inside the vesicle. Protons should permeate the channel (correct?) but the kinetic response would likely be different (delayed) from those receptors with their ECDs facing the solution exchange directly. Reconstitution was performed with labeled receptors, so fluorescence signal will be coming from both 'inward' and 'outward' oriented receptors. How does this complication affect analysis and interpretation of the time constants derived from multi-exponential fitting?

6) The concept of the intermediate state at intermediate pH, the pre-activation step, is compelling. One idea is that this intermediate state is a broadly-relevant point along the resting-activation pathway. What do the authors think of an alternative hypothesis, that the intermediate state results from partial occupancy of proton agonists at multiple sites that have different pKa values?

7) How does the locally closed model (Prevost NSMB 2012 and Gonzalez-Gutierrez PNAS 2013) fit into the concept of this new pre-activated conformation? Please also consider citing that 2013 PNAS paper.

---

## [Author Response]

*Essential revisions:*

*1) It is not clear whether the authors have conclusively ruled out the possibility that these effects on fluorescence quenching is not due to titration of di-amide group in bimane. I am not sure what the pKa of this group is but pKa of amide carbonyl is pH 3.0 and it is important to rule out the possibility that these changes in fluorescence are not due to protonation/deprotonation. For instance, one could argue that the differences in pH sensitivity just reflects the changes in pKa of the di-amide carbonyls due to the local environment rather than a conformational change. Although, the authors have shown that the fluorescence is unaffected going from pH 4.0 to pH 7.0, they cannot conclusively rule out the alternate possibility. This can be addressed by discussing these caveats when interpreting their data. In addition, the authors may want to show that the dye fluorescence is unaffected even the pH titration is extended by at least one pH unit on both ends.*

The referee is right to point out that this was not properly discussed in the text but only in the Methods section. We have now added the following text at the beginning of the Results section: “GLIC being activated by protons, we additionally performed control experiments to confirm that, as previous studies have shown (Jones Brunette and Farrens 2014), both the bimane fluorescence and its quenching by tryptophans are unaffected by proton concentrations ranging from pH 9 to pH 2 (Figure 3—figure supplement 1) (see Methods). Hence, the fluorescence variations of bimane-labeled mutants can be interpreted as reporting local structural reorganizations.”

As suggested by the referees, we extended our measurements of the fluorescence and tryptophan quenching of Bimane-Cys to pH 9 and pH 2 (Figure 3—figure supplement 1). The data show that both the fluorescence intensity and the quenching efficiency are unaffected, notably at a pH as low as 2. Clearly, the pH does not influence the bimane fluorescence and it is very likely that the bimane moiety is not protonated at any pH tested. This is consistent with its chemical structure. As amide carbonyl typically display pKa values lower than 0, it is reasonable to speculate that the pKa of the bimane imide carbonyls should be in the same negative range. In addition, simulations of pKa using ACDLab, ChemAxon or JChem base show that the bimane may be protonated at pHs lower than -9 on the remote hydrazide nitrogen. Moreover, the effects that are relevant to the GLIC allosteric transitions are essentially observed at pH 5 and pH 4, several orders of magnitude lower than the putative proton concentration necessary to protonate the bimane.

Lastly, we see that for quenching couples (e.g. Bimane-136 +/- W101 or Bimane-133 +/- W103) the bimane fluorescence is unaffected by pH variations up to pH 3 in absence of quencher (see Figure 3). The differences in bimane fluorescence are solely recorded when a tryptophan is at a quenching distance from the bimane, and thus the fluorescence variations discussed in the manuscript are due to the presence of a tryptophan and not to the protonation/deprotonation of the bimane.

*2) It is nice to see that the 243 position shows additional slow kinetics that is in the same range as channel opening but actual change compared to the large baseline noise is pretty small. Can the authors rule out alternate possibility that this small decrease is not due to bleaching, for instance? The fact that the pH titration curves at 243 position for TRIQ and Tl+ flux overlaps, is a strong evidence that this position probably reports on channel opening.*

In order to assess the possibility of fluorophore bleaching during stopped-flow experiments (30 s), we systematically performed control recordings of every mutant at pH 8, a pH for which there are no protein reorganizations. For most mutants, including the mutant Bimane-243 (shown in Figure 6—figure supplement 1), these recordings do not show fluorescence variation over the 30 s, indicating there is no significant bleaching of the bimane.

In addition, to be thorough and remove from the data any minor participation of fluorescence bleaching to the recorded variations, all the recordings performed at lower pHs were normalized to the pH 8 trace, as described in the Methods: “Each data set at pH 6, pH 5 and pH 4 was normalized to the integral 30 s recording made at pH 8 on the same day and same batch of proteoliposomes, to correct potential unspecific drifting/bleaching of fluorescence during the recordings.” (This protocol was applied for all mutants).

We wish to point out though, that the dose-response curves presented in Figure 7 for mutant 243 represent in black the fluorescence variations and in red the oocyte-recorded electrophysiological response rather than the thallium flux. It is likely that, as observed at position 136, the thallium flux dose-response curve would be slightly shifted to lower pHs as compared to the electrophysiological response in oocytes. This speculation would imply that the ΔF curve would be left-shifted as compared to ΔI curve, in agreement with the stopped-flow recordings that suggest that position 243 follows reorganizations both prior (very-fast component) and during (slower components) activation.

*3) Introduction, last paragraph: quenches the fluorophore when the distance is less than… Could you expand on that? Once you get below 15 Å for Trp, is there a relation between the effectiveness of quenching and the distance? For instance – D136 is always quenched by W93. This suggests that their distance is less than 15 Å in all conditions, but does it exclude this distance changing during activation, while remaining below 15?*

For molecules in solution with free diffusion/rotation, the relationship between the degree of quenching and the donor-acceptor distance in electron transfer mechanisms (both photoinduced or Dexter) is exponential (Dexter D.L., J Chem Phys 21, 836, 1953 and Strauch S. et al., J Phys Chem, 87, 3579 1983). However, when they are tied to a structured protein, the rate-distance relationship cannot simply be described by an exponential as it depends on the orientation of the fluorophore (Williams R.M. Photochem Photobiol Sci, 2010). This complexity, and thus the need for approximations in solving quenching equations, is the main reason why we never extrapolate the fluorescence variations to numerical values of distances. To evaluate distances changes, we used the empirical values collected by Jones Brunette and Farrens (2014) on the lysozyme, where tryptophan and tyrosine quench the fluorophore when the donor-acceptor Cα-Cα distances are less than approximately 15 Å and 10 Å, respectively, as stated in the Introduction.

In the specific case of the mutant Bimane-136-W93, the data indeed suggest that both the fluorophore and quencher are always at quenching distances from pH 7.3 to pH 3. There is a small variation between pH 7.3 and pH 6 that suggests a significant, although minor, distance variation. The signal is then stable from pH 6 to 3, and approximately corresponds to an 80% of fluorescence quenching. Thus, the quenching is not complete, since the theoretical maximal quenching is 100% when the fluorophore and quencher are always in contact. The stable 80% quenching can thus be interpreted in two ways: 1) there are no allosteric reorganizations occurring in this region from pH 6 to pH 3 or, 2) the region does undergo structural reorganizations but, on average, the mean quenching, that is the mean fluorophore/quencher distance of all the GLIC molecules in solution, is constant. The present data cannot discriminate between these two possibilities.

*4) Steady state quenching measurements at pH values < 5 are likely reporting on a desensitized conformation, correct? The results in the steady-state sections are framed largely from a context of understanding activation, but at low pH, from studies on patches, I would expect the receptor to be desensitized. Clarification of this point in a rebuttal and/or in the text (e.g. see subsection “The six fluorescent sensors report allosteric reorganizations of GLIC”, first paragraph) would be helpful to me.*

As stated in the Discussion, the desensitization of GLIC is not well understood, as illustrated by the patch clamp study of Laha et al. (2013) that shows that only half of the patches show macroscopic desensitization of the pH-elicited response. It is thus evident that in steady state conditions, fluorescence recordings are following reorganizations from the resting state towards the active and desensitized states, but we do not know the proportion of each of these states at the end of the equilibration. This is why in the Results section, we were careful to interpret the data in terms of pH-elicited reorganization but without referring to activation and desensitization.

We only address these questions in the last section of the Results and in the Discussion, as it is the combination of steady state and kinetics experiments that gives a coherent picture and allows us to discriminate between activation and desensitization.

At all positions except 243, we show that most of the fluorescence variations occurred with very fast kinetics, compatible with their contribution to the activation transition, particularly with regards to the known kinetics of GLIC published in the literature. We then conclude that the remaining minor quenching changes are linked to the transition from the active to the desensitized state.

These considerations, which are based on the entire set of experimental data, are stated in the Discussion, and could thus not be introduced earlier in the body of the Results section.

*5) Presumably, in liposome reconstitution experiments beginning at the subsection “Detergent-solubilized and lipid-reconstituted GLIC show similar pH-elicited reorganizations”, ~50% of the receptors have their ECDs (and H^+^ binding sites) inside the vesicle. Protons should permeate the channel (correct?) but the kinetic response would likely be different (delayed) from those receptors with their ECDs facing the solution exchange directly. Reconstitution was performed with labeled receptors, so fluorescence signal will be coming from both 'inward' and 'outward' oriented receptors. How does this complication affect analysis and interpretation of the time constants derived from multi-exponential fitting?*

The question of the referee is quite pertinent. As stated in the Methods section, we used two different reconstitution protocols for the thallium fluxes and the bimane-quenching experiments, yielding two types of liposomes with the same lipidic composition but large (150 nm in diameter) and small (70 nm in diameter), respectively.

In the thallium flux experiments, the ANTS fluorescence variations report on the fraction of open-channel receptors at the membrane, regardless of their orientation. We used these measurements not for the absolute value of thallium rate entry but to compare the fraction of active receptors between different pHs. In this frame, as the orientation of GLIC pentamers in the liposomes would be identical at all pHs, the possibility of delayed activation of receptors would not impact our conclusions.

However, in the case of the stopped-flow recordings of bimane, a delayed activation of GLIC could have a strong impact on the analysis and conclusions drawn. To address this issue, we performed three sets of experiments:

1) We evaluated the orientation of GLIC into liposomes. To this end, we reconstituted separately two unlabeled GLIC mutants for which a cysteine was introduced either at the top of the ECD (D136C), or at the bottom of the TMD facing the cytoplasmic side (T219C). After reconstitution, the proteoliposomes were labeled with the non-permeant fluorescent dye Alexa Fluor 546 maleimide for 10 minutes and the reaction was then stopped by addition of excess cysteines. The samples were subjected to SDS-PAGE, and gels were successively imaged by fluorescence and analyzed by Coomassie blue staining. In parallel, an identical labeling protocol was performed on the same DDM-solubilized mutants to measure the maximal labeling efficiency of each position. By analysis of each band’s intensity in fluorescence and Coomassie blue, we could calculate the fraction of labeled receptors (Alexa fluorescence intensity/Coomassie blue intensity) of both mutants. The data in the author response Figure 1 shows that, as compared to DDM-solubilized protein, 75 ± 11% of the D136C mutant and 35 ± 6% of the T219C mutant are labeled in liposomes, meaning that their mutated cysteine is facing the outside of liposomes. Our results thus strongly suggest a preferential orientation of GLIC with the ECD outside of the liposomes.

2) We measured the kinetics of proton entry into the liposome using stopped-flow recordings with identical injection volume and speed parameters as the ones used for bimane fluorescence recordings. We generated asolectin liposomes in the absence or presence of GLIC, which were loaded with the fluorescent pH-sensitive dye 8-HydroxyPyrene-1,3,6-Trisulfonic Acid (HPTS). When excited at 372 nm this dye shows a 1.7 fold increase in fluorescence intensity from pH 7.3 to pH 4 (Figure 9). Rapid mixing experiments from pH 7.3 to pH 4 show a slow proton entry into the liposomes (without GLIC), with time constants in the second to tens of seconds range (Figure 9). In sharp contrast, similar experiments with proteoliposomes (with GLIC) show very fast proton entry, which occurs with sub-millisecond kinetics, in the dead-time of the apparatus. As a control, we performed additional recordings under conditions for which GLIC is non-conducting, either of proteoliposomes going from pH 7.3 to pH 6 (pH too high for channel opening), or going from pH 7.3 to pH 4 in presence of the pore blocker picrotoxin. These recordings showed similar very-fast entry of the proton in the proteoliposomes indicating that this entry does not require GLIC-channel opening.

3) We evaluated the radius of the liposomes using dynamic light scattering. The data shows that the liposomes display a mean radius of approximately 70 nm, while proteoliposomes are smaller, with a mean radius around 35 nm (Figure 9). These data also show that our samples do contain formed mono-dispersed proteoliposomes and not individual lipidated GLIC particles.

In conclusion, we show that a non-negligible portion of the reconstituted proteins have an “inside-out” configuration with their ECD facing the inside of the vesicles. However, the proteoliposomes are permeant to protons with very fast kinetics. Therefore, in our recording conditions, the inner solution of the proteoliposomes is entirely equilibrated with the outer solution, in terms of proton concentration, within the mixing dead-time. Thus, all receptors, regardless of their orientation in the liposomes, will be protonated with sufficiently fast kinetics such that there is no impact on the observed bimane stopped-flow kinetics at the current time resolution.

Author response image 1.Characteristics of proteoliposomes used for Bimane quenching studies.(**A**) SDS-PAGE gel fluorescence scan (left) and labeling quantification (right) of GLIC mutants D136C and T219C reconstituted in liposomes (PL) or in detergent (DDM) and labeled with Alexa Fluor 546. (**B**) Dynamic light scattering profile of GLIC proteoliposomes. The peaks are represented with the standard radius-weighted intensity and the radius on a log scale. The first peak represents GLIC proteoliposomes (which are the major species of objects) and the second peak represents aggregates (very minor species). (**C**) Emission spectra of the HPTS fluorophore in proteoliposome buffer, excited at 372 nm at pH 7.3 and pH 4. (**D**) Stopped-flow recordings of HTPS loaded GLIC proteoliposomes (left) or empty liposomes (right). The lower left panel shows a zoom of the first second of the GLIC proteoliposomes recordings, represented using a log scale for better visualization of the first milliseconds. For all recordings, the excitation was set at 372 nm and the fluorescence recorded through a 505 nm high-pass filter after a 2.1 ms mixing dead-time to reach either pH 7.3 or pH 4.**DOI:**
http://dx.doi.org/10.7554/eLife.23955.026

*6) The concept of the intermediate state at intermediate pH, the pre-activation step, is compelling. One idea is that this intermediate state is a broadly-relevant point along the resting-activation pathway. What do the authors think of an alternative hypothesis, that the intermediate state results from partial occupancy of proton agonists at multiple sites that have different pKa values?*

The question is interesting and quite pertinent, but in light of the present data we can only speculate on alternative models. A number of different mechanisms can account for the fluorescence data, and two have been proposed by the referees:

1) The simplest one is described in the “Additional Note by one of the reviewers”. In this case, a single class of “proton activation site” is present, one per subunit, five per pentamer. As shown by the theoretical curves provided by the referee, the concerted model of R to pre-O to O nicely accounts for the experimentally measured dose response curves, notably for the left shift of the △F curves as compared to the △I curve.

2) It is also possible that more than one “proton activation site” per subunit exists. Each subunit could carry a low sensitivity and a high sensitivity site, and it is possible that the transition towards the pre-active state would reflect the protonation of the high sensitivity sites, the activation requiring further protonation of the low sensitivity sites.

Future work, notably by mutagenesis, would be required to discriminate between these possibilities.

*7) How does the locally closed model (Prevost NSMB 2012 and Gonzalez-Gutierrez PNAS 2013) fit into the concept of this new pre-activated conformation? Please also consider citing that 2013 PNAS paper.*

We added a section (Discussion, fourth paragraph) discussing the possibility that one particular LC conformation could be a candidate for the pre-active state. This remains highly speculative and is stated as such in the text: “Several mutants and a wild-type GLIC C-terminally tagged with 10 histidines, were found to crystallize in different locally-closed “LC” conformations characterized by an overall conformation of the ECD nearly identical to GLIC_pH 4_, but with a closed channel (Prevost et al. 2012; Gonzalez-Gutierrez et al. 2013). […] Hence, we may speculate that this particular LC conformation could be a candidate for the pre-active state of GLIC, since it shows compaction of the ECD and motion of the M2-M3 loop without channel opening.”

MINOR POINTS:

1. How do the authors rule out that the ion influx kinetics is not limited by the number of functional channels? There should be some discussion about this. Please provide information regarding the size of the liposomes, number of dye molecules per liposome etc.

The ion flux kinetics are dependent on the number of functional/active channels as documented in previous reports establishing this technique (Ingolfsson and Andersen 2010). The amount of protein used for the reconstitution thus has to be adjusted to yield quenching kinetics that can be resolved at the highest and the lowest channel activity under the specific conditions of the experiment. Since all the experiments comparing kinetics under different activating conditions are measured using vesicles from the same reconstitution, the differences in kinetics can be safely assigned to differences in the pH dependent activation. Moreover, in the experiments shown here, the recorded channel activity from different reconstitutions shows a variation of quenching rates of less than 10%. This point is now addressed in the method section for the flux assay p.29 l27-28.The LUVs prepared according to this protocol are extruded through a 100 nm filter and have a mean diameter of 150 nm after extrusion (Ingolfsson and Andersen 2010). This is now specifically stated p.19 l14. Thus the mean volume of the vesicles is 1.7671 mm^3^ ·10^-12^ = 1.767x10^-12^ ml. With 25 mM ANTS present during the reconstitution (see Methods p.29), the actual number of ANTS molecules inside the vesicles is 4.4175x10^-14^ mol or 2.66x10^10^ molecules. In every experiment shown here, fluorescence is quenched to about 70% of the initial fluorescence (under these particular conditions fluorescence quenching cannot decrease to less than 40% of the initial fluorescence (Ingolfsson and Andersen 2010)). Thus the amount of ANTS molecules is not limiting for the quenching reaction.

2. 100% labeling efficiency seems unlikely. In cases where labeling results in a non-functional channel, is there a concern that some experiments are measuring function from the fraction of receptors that are unlabeled?

The referee is right to point out this issue. First, for mutations at positions E243 or P250, we found that bimane labeling altered the pH50 of activation and/or the shape of the current traces. In these cases, as there is a difference in the currents, electrophysiology clearly does record labeled receptors.In contrast, at position 133, 135, 136 and 33, there is no such significant change in the pH50 or current kinetics observed after bimane labeling. In addition, in all cases bimane labeling did not decrease the maximal currents evoked by proton (see Table 1). Since positions 133, 135 and 136 are far away from the transmembrane domain, it is likely that their labeling with bimane is similar between the membrane-anchored protein and the DDM-solubilized protein. We thus evaluated the extent of bimane labeling in DDM-purified protein (Table 6). To this end, we calculated the efficacy of bimane labeling by dividing the fluorescence of the bimane-labeled proteins in denatured condition (SDS) by the amount of protein. This F_SDS_/protein ratio calculated for all the mutants indicates that the position Q193C has the lowest labeling efficiency (37 units of fluorescence per mg of receptor), in particular as compared to the major quenching pairs D136C Q101W (85 units) V135C (119 units), R133C L103W (53 units) and K33C (113 units). Functional recordings of the mutant Q193C shows that our labeling protocol produces almost 100% of function loss. Therefore, we can extrapolate that at all other positions, as the labeling efficiency is higher, if bimane labeling were to lead to a complete loss of function we would, as for Q193C, record it in oocytes. We conclude that, in these cases, bimane labeling did not significantly impair the function of the channel.

Author response table 1.Labeling efficiency of GLIC mutants by Bimane. n stands for number of experiments. All the data represent mean values and error values were calculated as standard deviations.**DOI:**
http://dx.doi.org/10.7554/eLife.23955.027MutantF_SDS_/proteinnK33C113 ± 414K33C W160F54 ± 413D136C155 ± 353D136C S93W522D136C D178W106 ± 43D136C Q101W85 ± 144R133C60 ± 44R133C Y23W482R133C Q101W60 ± 83R133C L103W53 ± 74V135C119 ± 367V135C E67Q E75Q D91N108.6 ± 0.63 Q193C 37 ± 33P250C70 ± 154P250C W160F70.9 ± 0.63P250C Y194F52.6 ± 0.4 3 P250C Y197F60 ± 43P250C Y251F68.1 ± 0.63E243C113 ± 24E243C K33W102 ± 23E243C I201W118 ± 53E243C F238W69.7 ± 0.73E243C V242C16 ± 13E243C L241W101 ± 73